# UNICO: ON UNIFIED COMBINATORIAL OPTIMIZATION VIA PROBLEM REDUCTION TO MATRIX-ENCODED GENERAL TSP

**Wenzheng Pan**[†]**, Hao Xiong**[†]**, Jiale Ma, Wentao Zhao, Yang Li, Junchi Yan**[*]
Sch. of Computer Science & Sch. of Artificial Intelligence, Shanghai Jiao Tong University
{pwz1121,taxuexh,heatingma,permanent,yanglily,yanjunchi}@sjtu.edu.cn
Code: https://github.com/Thinklab-SJTU/UniCO

## ABSTRACT

Various neural solvers have been devised for combinatorial optimization (CO), which are often tailored for specific problem types, e.g. TSP, CVRP and SAT, etc. Yet, it remains an open question how to achieve universality regarding problem representing and learning with a general framework. This paper first proposes **UniCO**, to unify a set of CO problems by reducing them into the *general* TSP form featured by distance matrices. The applicability of this strategy depends on the efficiency of the problem reduction and solution transition procedures, which we show that at least ATSP, HCP, and SAT are readily feasible. The hope is to allow for the effective and even simultaneous use of as many types of CO instances as possible to train a neural TSP solver, and optionally finetune it for specific problem types. In particular, unlike the prevalent TSP benchmarks based on Euclidean instances with 2-D coordinates, our studied domain of TSP could involve non-metric, asymmetric or discrete distances without explicit node coordinates, which is much less explored in TSP literature while poses new intellectual challenges. Along this direction, we devise two neural TSP solvers with and without supervision to conquer such matrix-formulated input, respectively: 1) **MatPOENet** and 2) **MatDIFFNet**. The former is a reinforcement learning-based sequential model with pseudo one-hot embedding (POE) scheme; and the latter is a Diffusion-based generative model with the mix-noised reference mapping scheme. Experiments on ATSP, 2DTSP, HCP- and SAT-distributed general TSPs show the strong ability towards arbitrary matrix-encoded TSP with structure and size variation.

## 1 INTRODUCTION

Beyond heuristics, learning-based neural solvers have shown success in solving combinatorial optimization (CO) problems. While designing neural solvers for a specific type, e.g., TSP, has been a popular pursuit (Khalil et al., 2017; Vinyals et al., 2015), yet such a problem type-specific paradigm can be restrictive in real-world with immense problem diversity, which prompts the following question:

- *Can we develop a **general framework** capable of learning a range of CO problems on graph in a unified manner?*

In this paper, we resort to utilizing **problem reduction**, namely transformation between different problem types which is in fact largely neglected by previous research. By the theory of computational complexity, any NP problem can be transformed (or in a more professional terminology, *reduced*) into an NP-Complete (NPC) problem in polynomial time, and every NP problem can be reduced to an NP-hard problem in polynomial time (Lewis, 1983; Van Leeuwen, 1991). The famous Karp's 21 NPC problems (Karp, 1972) exemplify the polynomial-time reduction between 21 NPC CO problems.

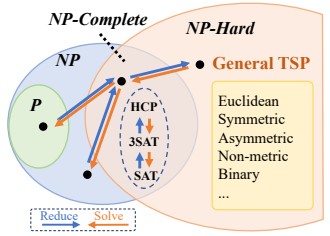

Figure 1: CO problems and the polynomial-time reduction. HCP, 3SAT and SAT are used as case studies in this paper.

---
[*]Corresponding author, who is also affiliated with Shanghai Innovation Institute. [†]Equal contribution, in alphabetical order of the last name. This work was partly supported by NSFC (92370201, 62222607).

Table 1: Comparison of different current works with the key word "multi-task solver". Refer to Table 6 in Appendix B.4 for a more thorough comparison of recent related works.

| Method | MTCO (Li & Liu, 2023) | ASP (Wang et al., 2024) | MAB-MTL (Wang & Yu, 2023) | MVMoE (Zhou et al., 2024) | UniCO (Ours) |
|---|---|---|---|---|---|
| Evaluated Problems | PFSP | 2DTSP, CVRP | TSP, CVRP, OP, KP | VRPs | ATSP, 2DTSP, DHCP, 3SAT |
| Applicable Problems | | N/A (no solver is proposed) | | | {problem $\mathcal{P}$ \| $\mathcal{P} \leq_P$ general TSP} |
| Multi-Task | ✗ | | ✓ | Limited to VRP variants | ✓ |
| Multi-Scale | ✓ | | ✓ | ✓ | ✓ |
| Single Solver | ✗ | | ✗ | ✓ | ✓ |
| Solver Type | Quadratic Programming | Evaluated on ML-based ones | ML-based | ML-based | ML-based |
| Brief Description | Enhance PFSP solving via knowledge transferring between similar instances | A model-/problem-agnostic training framework to improve generalizability | A multi-task neural solver for specific problems via multi-armed bandits | A multi-variant VRP solver with hierarchical gating mechanism | A neural CO framework with two solvers for problems reducible to general TSP |

Specifically, by introducing the general Traveling Salesman Problem (general TSP, see Def. 1) as the reduction endpoint, one can construct a problem reduction tree as shown in Fig. 1, suggesting the potential of training a general TSP solver to tackle problems within the reduction tree. While theoretically, other NP-hard problems, e.g., mixed integer linear programming (MILP), could also serve as the reduction endpoint (Zhang et al., 2023), TSP has already fostered extensive research attention in neural solvers in recent literature (U et al., 2021). In fact, existing machine learning research on (end-to-end) MILP neural networks are sill limited (Zhang et al., 2023), compared with the emerging progress in neural TSP (Li et al., 2023b; Ye et al., 2024b; Drakulic et al., 2023) solvers.

In the context of developing a general neural TSP solver for various CO problems as discussed above, the reduced TSP instances in fact can be represented by an arbitrary distance matrix in both its size as well as its elementary value, far beyond the popular 2D points form (Kwon et al., 2020) in the well-studied Euclidean space. In another word, it can be no longer a metric space and the distance may break the triangular inequality. Currently, Transformer (Vaswani et al., 2017) and graph neural network (GNN) have become popular backbones for neural TSP solvers (Joshi et al., 2019; Kool et al., 2018) where the 2D coordinates of the points are often used as node features, which constraints the research scope of the TSP world. For the general TSP with arbitrary matrix as input, it is technically nontrivial to achieve effective featuring of the problem instances with only pair-wise relationships and have non-deterministic node numbers (Joshi et al., 2020). To address such challenges, this paper proposes a novel framework called UniCO and two corresponding neural solvers (MatPOENet and MatDIFFNet) that innovatively improves prevalent Transformer and Diffusion models as matrix encoder for general TSP learning and solving. **Our contributions can be summarised as follows:**

- We conceptualize **UniCO**, namely Unified Combinatorial Optimization learning framework, leveraging the rich expressivity of general TSP with arbitrary positive-valued matrix for unified representation of multiple CO problems (where reducible). We also construct standard datasets benchmarking the under-explored capacity of the general TSP world accordingly. This practice, to our best knowledge, has not been performed, especially in the context of machine learning for CO.
- We propose **MatPOENet**, namely Matrix encoding Network with Pseudo One-hot Embedding, a reinforced Transformer-based model which utilizes a novel size-agnostic node embedding to aid instance input, thereby significantly improving model scalability and performance of general TSP.
- We propose **MatDIFFNet**, namely Matrix encoding Diffusion Network, a supervised diffusion-based model which leverages a novel mix-noised reference map module, thus extending the promising ability of generative model for Euclidean TSP solving to matrix-formulated general TSP.
- We instantiate UniCO with the above two proposed neural backbones and one more existing method, DIMES, and conduct experiments on general TSP with four types of CO problem distributions, i.e., ATSP, 2DTSP, DHCP, and 3SAT. Experiments show that measuring either the average TSP tour length or the average rate that solvers find optimal solutions for decisive tasks, our best-performing methods beat compared neural approaches, and outperform the strong heuristic LKH in some cases.

## 2 RELATED WORKS

**Multi-Task CO Solvers.** To the best of our knowledge, this is the first work that combines neurally matrix-encoded general TSP solver and unified multi-task CO learning. While similar concepts, e.g., "multi-task solver", "universal solver", etc., have also appeared in recent literature, they often denote quite different approaches and functionalities. For example, **MTCO** (Li & Liu, 2023) devises a "multi-task" CO framework which measures the similarity between CO problems, and then transfers knowledge between similar instances within the same problem type to gain search speed-ups in its quadratic programming solver. However, it solely focuses on the permutation flowshop scheduling problem (PFSP) and is not specifically designed for learning-based neural solvers, making the

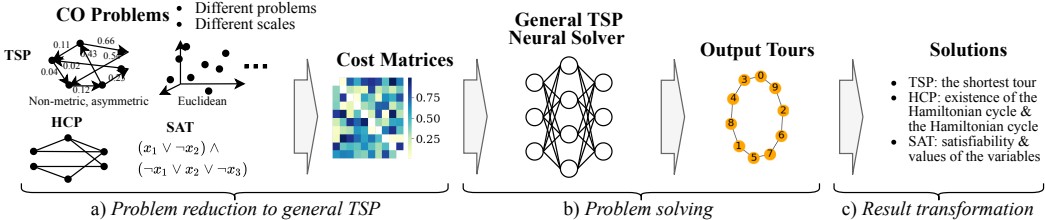

Figure 2: The 3-step workflow of the UniCO learning framework.

term "multi-task" in its title misleading from a learning perspective and irrelevant to the theme of this paper. **ASP** (Wang et al., 2024) proposes a "universal" framework to address generalization issues of neural CO solvers with a model-/problem-agnostic training policy. However, no specific new solver is proposed, which we believe is technically nontrivial to deal with different problems. **MAB-MTL** (Wang & Yu, 2023) proposes a multi-armed bandit framework to train a neural solver with a shared encoder but different header and decoder. Yet, as the header and decoder necessitate customized designs for specific problems, the solver cannot be readily applied to unseen problems beyond its four training tasks. **GCNCO** (Li et al., 2025a) builds upon the similar header-encoder-decoder structure yet additionally enforces the consistency of the optimization trajectories across different problems to promote learning generalizable strategies corresponding to the shared structure among different CO problems. **MVMoE** (Zhou et al., 2024) proposes a "multi-task" vehicle routing solver with mixture-of-experts and a hierarchical gating mechanism to enhance the model capacity with good computational complexity. However, its applicable problems are also limited to variants of VRPs. Table 1 outlines the disparities between representative existing works and ours, highlighting the uniqueness and novelty of our work. More recent works like **MTNCO** (Liu et al., 2024), **UNCO** (Jiang et al., 2024), and **GOAL** (Drakulic et al., 2024), etc., are discussed in Appendix B.4.

**General TSP Solvers.** Besides traditional solvers and heuristics, neural methods for TSP with 2D coordinates have been well studied in literature (Kool et al., 2018; Kwon et al., 2020; Qiu et al., 2022; Sun & Yang, 2023; Li et al., 2023b; 2024; Xin et al., 2021), as systematically discussed in Li et al. (2025b). However, neural solvers for general TSP are much more challenging due to a lack of effective and scalable neural networks to handle the pairwise distance information. To our best knowledge, few works have demonstrated comparable capability of general TSP solving. **MatNet** (Kwon et al., 2021) proposes a Transformer-based solver for asymmetric TSP (ATSP), which takes distance matrices as input. However, the adopted one-hot embedding has limitations in dealing with arbitrarily large matrices, and its implementation typically includes a preset value of maximum size, restricting its generalization ability. **BQ-NCO** (Drakulic et al., 2023) proposes leveraging bisimulation quotienting to enhance out-of-distribution robustness on CO problems, which is capable of solving ATSP via Markov Decision Process (MDP) formulation. The following research lines show the potential of adaptation to general TSP solving without major revisions to the model architecture or learning paradigm. Divide-and-conquer methods (Ye et al., 2024b; Zheng et al., 2024b; Luo et al., 2023) generally learn to break down a problem into sub-problems to facilitate solving, which accommodate matrix encoding models like MatNet as sub-problem solvers for general TSP solving. Non-autoregressive methods, including predictive solvers (Joshi et al., 2019; Qiu et al., 2022; Min et al., 2024) and recent generative solvers (Sun & Yang, 2023; Li et al., 2023b; 2024), typically utilize GNNs to encode both node features (coordinates) and edge features (distances) for directly predicting solutions, which can be adapted for general TSP solving with minor modifications to the node features at the cost of subpar results.

In Appendix B, we further discuss exact solvers, heuristic solvers for general TSP; auto- and non-auto-regressive neural solvers and neural-heuristic solvers for 2DTSP, divide-and-conquer methods for large routing problems, and specific solvers for other covered CO problems (HCP, SAT, etc.).

## 3 PRELIMINARIES

### 3.1 COVERED CO PROBLEMS

**Definition 1** (Traveling Salesman Problem (TSP)). *Given a complete, directed or undirected graph without self-loops denoted by $G = (\mathcal{V}, \mathcal{E})$ ($\mathcal{V} = \{1, 2, \cdots, N\}$: the node set, $\mathcal{E}$: the edge set) along with a cost matrix $\mathbf{C}$ of the shape $N \times N$ where the entry $\mathbf{C}_{ij}$ is the cost for edge $(i, j) \in \mathcal{E}$, the problem is to find the tour $\tau = (i_1, \cdots, i_N)$ to minimize the total cost $\sum_{k=1}^{N-1} \mathbf{C}_{i_k i_{k+1}} + \mathbf{C}_{i_N i_1}$.*

---

**Algorithm 1** The training pipeline of UniCO.

---

**Input:** Problems $\{\mathcal{P}\}$ reduced to general TSP, batch size $B$, a general TSP solver $\mathcal{S}$ handling instances of different scales.

**repeat**

    Select a problem from $\{\mathcal{P}\}$ and generate a batch of $B$ instances of the problem at the same scale;

    Reduce the instances to general TSP instances with cost matrices $\{\mathbf{C}^{(b)}\}_{b=1}^B$;

    Train $\mathcal{S}$ with the input instances $\{\mathbf{C}^{(b)}\}_{b=1}^B$;

**until** the training of $\mathcal{S}$ converges;

---

**Definition 2** (**Hamiltonian Cycle Problem (HCP)**). *Given a directed or undirected graph $G = (\mathcal{V}, \mathcal{E})$, the problem is to determine whether there exists a Hamiltonian cycle in $G$.*

**Definition 3** (**Boolean Satisfiability Problem (SAT) in conjunctive normal form (CNF)**). *SAT aims to determine the existence of an interpretation that satisfies a given Boolean formula. A Boolean formula in CNF is represented by a conjunction (denoted by $\wedge$) of clauses that are disjunctions (denoted by $\vee$) of variables. For example, $(x_1 \vee \neg x_2) \wedge (\neg x_1 \vee x_2 \vee \neg x_3)$ is a SAT instance of two clauses $(x_1 \vee \neg x_2)$ and $(\neg x_1 \vee x_2 \vee \neg x_3)$, and three variables $x_1$, $x_2$ and $x_3$.*

Special cases of TSP are defined according to the properties of the cost matrix $\mathbf{C}$:

- *Metric.* A TSP is metric if the triangle inequality, i.e., $\mathbf{C}_{ij} + \mathbf{C}_{jk} \geq \mathbf{C}_{ik}$, holds for any different three nodes $i$, $j$, and $k$. Specially, when $\mathbf{C}$ is derived from coordinates in Euclidean space, the TSP is *Euclidean*.
- *Symmetric.* A TSP is symmetric if $\mathbf{C}_{ij} = \mathbf{C}_{ji}$ for all $i$ and $j$; otherwise, it is *asymmetric*.

We use the term "*general TSP*" to refer to TSPs either metric or non-metric, symmetric or asymmetric. For HCP, without losing generality, we mainly discuss directed HCP (DHCP) in this paper. For SAT, a special case of SAT in CNF with at most 3 variables is named as 3-Satisfiability Problem (3SAT). Note any SAT problem can be reduced to 3SAT in polynomial time (Fouh et al., 2014). Throughout this paper, for consistent representation and unified evaluation, when we mention any problem type X, we prescribe a limit to its reduced general TSP formulation, i.e., differently distributed distance matrices only. Furthermore, note that a wider range of problems (e.g., VC, Clique, certain VRPs, FFSP, MIS, etc., as detailed in Appendix E.2), can be dealt with by our proposed methods.

### 3.2 Polynomial-Time Reduction of CO to General TSP

The reduction relations of the covered CO problems in this paper can be summarized as 3SAT $\leq_P$ HCP $\leq_P$ TSP ($A \leq_P B$ means that the problem $A$ can be reduced to $B$ in polynomial time). The solutions can also be transformed from a TSP tour to the solution of the raw problem. The detailed illustration of instance reduction and proofs are provided in Appendix A.

## 4 Methodology

### 4.1 UniCO: Unified CO Learning Framework

As illustrated in Fig. 2, given a CO instance, UniCO works in a 3-step pipeline: a) reduce the instance to a general TSP instance with a distance matrix obtained by techniques in Sec. 3.2, b) feed the distance matrix to a trained general tsp solver and output a tour, and c) transform the output tour into the solution of the origin problem (efficiently). To prepare the training data, we build a problem pool with CO problems of different scales that can be reduced to general TSP (see Sec. 5.1 for details). During training, we randomly fetch problems from the pool, transform them to general TSP, and treat the general CO solver training as an equivalent task to train the general TSP solver. A concise pipeline is given in Algorithm 1. In the main context, we propose increments to two promising architectures in solving general TSP across scales: the Transformer-based MatPOENet, and the Diffusion-based MatDIFFNet, trained in the reinforcement (Williams, 1992) and supervised manner, respectively. We describe the two architectures in Sec. 4.2 and Sec. 4.3. To demonstrate the general applicability of UniCO to different backbone solvers, we additionally incorporate DIMES (Qiu et al., 2022), a neural model based on GNN and meta-reinforcement learning, into our UniCO framework. Details of model, training, and results of UniCO-DIMES are presented in Appendix E.1.

### 4.2 Transformer-based Solver: MatPOENet

**Overview.** As is known, MatNet (Kwon et al., 2021) is the first neural solver designed for matrix encoding and asymmetric TSP. To overcome the inherent drawback that vanilla MatNet cannot scale

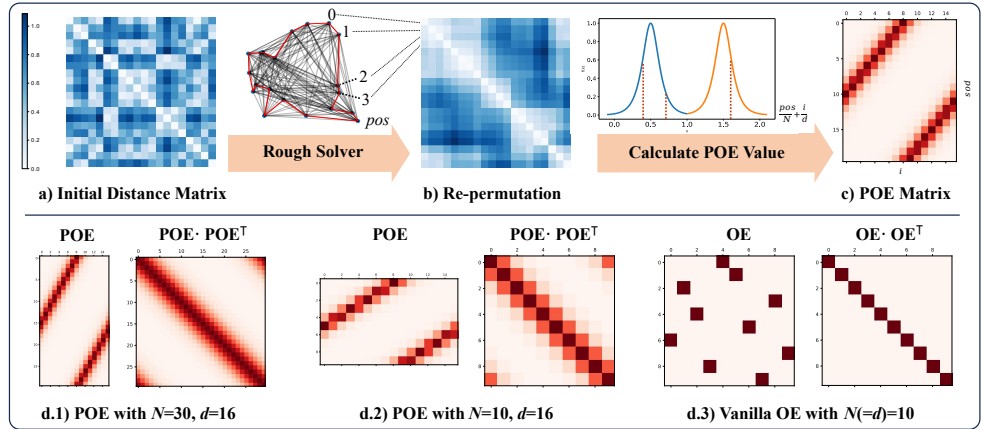

Figure 3: a-c) The three steps to generate the pseudo one-hot embedding (POE). d) The visualization of POE when $N > d$ (d.1) and $N < d$ (d.2) and one-hot embeddings (OE) (d.3), lighter blocks represent higher values. As the approximation of OE, POEs in Fig. d.1) and d.2) both achieve concentrating POE pairs of high dot products near the diagonal of the dot product matrix.

to arbitrary size of input due to its fixed initial node embeddings, we propose **MatPOENet**, namely Matrix encoding Network with Pseudo One-hot Embedding for general TSP, which greatly improves model scalability as well as performance. MatPOENet utilizes an encoder-decoder Transformer architecture to learn features of each city in the TSP, and is trained with deep reinforcement learning scheme (DRL). Generally, the model takes two sets of initial node embeddings $\mathbf{a}_0, \mathbf{b}_0$ and a distance matrix $\mathbf{C}$ as input, and sequentially calculates the probability of nodes selected for the next position in the TSP tour via standard attention operations and a mask indicating whether a node has been visited. The model is trained using POMO (Kwon et al., 2020) by DRL. For each instance with distance $\mathbf{C}$, $N$ tours $\{\tau^1, \cdots, \tau^N\}$ with different starting nodes are sampled to calculate the policy gradient:

$$\nabla_\theta J(\theta) \approx \frac{1}{N} \sum_{n=1}^{N} \left(L(\tau^n) - b(\mathbf{C})\right) \nabla_\theta \log p_\theta(\tau^n | \mathbf{C}), \tag{1}$$

where $L(\tau)$ is the length of tour $\tau$, $b(\mathbf{C}) = 1/N \sum_{n=1}^{N} L(\tau^n)$ is a baseline method, set as the mean tour length of the $N$ tours. Mathematical details of the model and training process is deferred to Appendix C.1 and the general network structure is correspondingly illustrated in Fig. 4. The POE technique introduced in Sec 4.2.1 is the highlight of our design, taking its effect by assigning $\mathbf{b}_0$ a non-trivial position embedding to largely elevate the performance and scalability of the model.

### 4.2.1 PSEUDO ONE-HOT EMBEDDING

In vanilla MatNet, zero embedding and one-hot embedding is adopted for initial node embedding. However, the one-hot component is of fixed size and cannot accommodate arbitrary input matrix sizes. Thus, we seek an input size-agnostic encoder with dimension $d$, thereby dismissing the need for the one-hot embedding. As a solution, we propose the Pseudo One-hot Embedding (POE) in the continuous vector space to replicate the functionality of the vanilla one-hot embedding. We denote the input embedding as $\mathbf{x}_i \in \mathbb{R}^d$ where $i$ is the node index. When we are applying the model on large instances, i.e., $N > d$, we can never ensure $\mathbf{x}_i \cdot \mathbf{x}_j = 0, \forall i \neq j$ as with the one-hot embedding. That means, there will unavoidably be some bias on the similarities between embeddings. So naturally, we incorporate this bias into the POE design by assigning similar embeddings to nodes that are likely to be connected in the solution.

Specifically, the POE works by the following steps: **1) (Fig. 3 a)** We transform the input cost matrix into a tour, forming a closed loop by a rough solver. To ensure efficiency, we adopt the *nearest neighbor* (NN) heuristic as the rough solver, which can also be replaced by other possible choices. Subsequently, each node on the tour is assigned a position index denoted by *pos* ranging from 0 to $N - 1$. **2) (Fig. 3 b)**: We define an even function $f : [-\frac{1}{2}, \frac{1}{2}] \to (0, 1)$ with an "impulse-like" shape, as depicted in Fig. 3 b). The POE is then generated based on $f$, with empirical success found using $f(x) = 1/\cosh(100x)$. **3) (Fig. 3 c).** We rotationally shift $f$ along the tour for different positions, and generate POEs by sampling values from $f$. Mathematically, the $i$-th entry of the POE for position

*pos* can be given by:

$$\mathbf{P}_{pos,i} = \begin{cases} f(\frac{pos}{N} + \frac{i}{d} - \frac{1}{2}) & \text{if } \frac{pos}{N} + \frac{i}{d} \le 1 \\ f(\frac{pos}{N} + \frac{i}{d} - \frac{3}{2}) & \text{if } \frac{pos}{N} + \frac{i}{d} > 1 \end{cases}. \tag{2}$$

The POE matrix $\mathbf{P}$ then becomes the substitution of $\mathbf{B}^0$ in the vanilla MatNet. To visualize the similarity between POE and the original one-hot embeddings, Fig. 3 d.1-3) shows different cases of the POE and one-hot embeddings.

### 4.3 DIFFUSION-BASED SOLVER: MATDIFFNET

**Overview.** Generative methods gain considerable attention and show promising performance in TSP solving. Prominent models for the generative objective encompass variational autoencoders (VAE) (Hottung et al., 2021a), diffusion models (e.g., DIFUSCO (Sun & Yang, 2023) and T2T (Li et al., 2023b)), and consistency models (Li et al., 2024). These representative works demonstrate good feasibility and competitive results of diffusion model solving TSP in the Euclidean space. However, a robust generative backbone for *general matrix-formulated* (A)TSP which finely adapts our UniCO framework has yet to be proposed. To fill this gap, we devise **MatDIFFNet**, namely Matrix encoding Diffusion Network for general TSP, which can also be seamlessly incorporated in our UniCO pipeline for multi-task and multi-scale unified training and solving. MatDIFFNet is inspired by and developed upon Sun & Yang (2023) and Li et al. (2023b), endeavoring to characterize a distribution of high-quality solutions for a given instance, i.e., estimating $p(\mathbf{S}|\mathbf{C})$, where $\mathbf{S}$ is the solution distribution and $\mathbf{C}$ the distribution of distance matrix. the general framework of diffusion includes a forward noising and a reverse denoising Markov process. The noising process takes the initial solution $\mathbf{S}_0$ and progressively introduces noise to generate a sequence of latent variables $\mathbf{S}_{1:T}$. The denoising process is learned by the model, which starts from the final latent variable $\mathbf{S}_T$ and denoises $\mathbf{S}_t$ at each time step to generate the preceding variables $\mathbf{S}_{t-1}$ based on the instance $\mathbf{C}$, eventually recovering the target data distribution. The formulation of the denoising process is expressed as $p_\theta(\mathbf{S}_{0:T}|\mathbf{C}) = p(\mathbf{S}_T) \prod_{t=1}^{T} p_\theta(\mathbf{S}_{t-1}|\mathbf{S}_t, \mathbf{C})$. The training optimization aims to align $p_\theta(\mathbf{S}_0|\mathbf{C})$ with the data distribution $q(\mathbf{S}_0|\mathbf{C})$ using ELBO:

$$\mathcal{L} = \mathbb{E}_q \left[ \sum_{t>1} D_{KL} \left[ q(\mathbf{S}_{t-1}|\mathbf{S}_t, \mathbf{S}_0) \parallel p_\theta(\mathbf{S}_{t-1}|\mathbf{S}_t, \mathbf{C}) \right] - \log p_\theta(\mathbf{S}_0|\mathbf{S}_1, \mathbf{C}) \right] + C. \tag{3}$$

We defer the mathematical elaboration of discrete diffusion process (derivation through Bayesian theorem, transition probability matrix, etc.) to Appendix C.2 and the general network structure is correspondingly illustrated in Fig. 5. The adaptive scheme proposed to enable matrix-input for our generative general TSP solver is detailed in the following Sec. 4.3.1.

### 4.3.1 MIX-NOISED REFERENCE MAP AND DUAL FEATURE CONVOLUTION

In previous works, graph-based diffusion networks for TSP takes two core inputs for the GNN encoder. One is the Euclidean 2D coordinates as initial node features, and the other is the noised reference map which forms the initial edge embedding. However, in general TSP formulation, no coordinates is available while and arbitrary distance matrix is instead provided. We endeavor to maintain the best compatibility with previous design principles, thus proposing to combine the distance matrix and the noised label matrix to obtain a mix-noised reference map to replace the original $\mathbf{x}_t$ to leverage edge information from pair-wise distances and enrich the initial edge embedding. Additionally considering the asymmetry of the distance matrix for general TSP and inspired by the scheme adopted in Kwon et al. (2021), we introduce two random vectors as pseudo coordinates for both "from" points and "to" points as node inputs, which will be updated respectively in subsequent GNN aggregations.

**Mix-Noised Reference Map.** The fusion of distance matrix $\mathbf{C} \in \mathbb{R}^{N \times N}$ and original noised reference map $\mathbf{x}_t \in \mathbb{R}^{N \times N}$ is learned by a multilayer perceptron (MLP) with two input nodes and a single output node and biases, conforming to that of vanilla MatNet. Mathematically, the mix-noised reference map $\mathbf{x}_t^{\mathbf{C}} \in \mathbb{R}^{N \times N}$ for the diffusion encoder can be calculated as follows. First, stack $\mathbf{C}$ and $\mathbf{x}_t$ along the last dimension to gain a mixture tensor $M \in \mathbb{R}^{N \times N \times 2}$, where $N$ denotes the number of nodes and the batch size is omitted. Then, linear and activating operations are performed on $M$:

$$\mathbf{x}_t^{\mathbf{C}} = W_{\text{mix2}} \left( \text{ReLU} \left( W_{\text{mix1}}(M) \right) \right). \tag{4}$$

Subsequently, $\mathbf{x}_t^\mathbf{C}$ is used to compute the initial edge embedding for the GNN via sinusoidal featuring of each input element respectively:

$$\tilde{e}_i = \text{concat}\left(\sin\frac{e_i}{T^{\frac{0}{d}}}, \cos\frac{e_i}{T^{\frac{0}{d}}}, \sin\frac{e_i}{T^{\frac{2}{d}}}, \cos\frac{e_i}{T^{\frac{2}{d}}}, \ldots, \sin\frac{e_i}{T^{\frac{d}{d}}}, \cos\frac{e_i}{T^{\frac{d}{d}}}\right), \quad (5)$$

where $e_i$ denotes the $i$-th value of the $N^2$ entries in $\mathbf{x}_t^\mathbf{C}$, $d$ is the embedding dimension, $T$ is a large number (usually selected as 10000), $\text{concat}(\cdot)$ denotes concatenation.

**Dual Feature Convolution.** Let $t^0 \in \mathbb{R}^{d_t}$, where $d_t$ is the time feature embedding dimension. $e$ is the mix-noised reference map calculated above. As for node features, deviating from the widely used GCN model (Joshi et al., 2019) that learns single node representation, we introduce $x_A, x_B \in \mathbb{R}^{N \times 2}$, two random generated pseudo coordinates as initial node embeddings for the asymetric nodes in general TSP, and maintain two distinct node features with two sets of learnable parameters throughout the cross-layer convolution operations:

$$x_{A,i}^{l+1} = x_{A,i}^l + \text{ReLU}(\text{BN}(W_{A,1}^l x_{A,i}^l + \sum_{j\sim i} G_{ij}^l \odot (W_{A,2}^l x_{A,j}^l + W_{B,2}^l x_{B,j}^l))), \quad (6)$$

$$x_{B,i}^{l+1} = x_{B,i}^l + \text{ReLU}(\text{BN}(W_{B,1}^l x_{B,i}^l + \sum_{j\sim i} G_{ij}^l \odot (W_{A,2}^l x_{A,j}^l + W_{B,2}^l x_{B,j}^l)^\top))), \quad (7)$$

$$e_{ij}^{l+1} = e_{ij}^l + \text{ReLU}(\text{BN}(W_3^l e_{ij}^l + W_{A,4}^l x_{A,i}^l + W_{B,4}^l x_{B,j}^l)) + W_5^l(\text{ReLU}(t^0)), \quad (8)$$

$$G_{A,ij}^l = \frac{\sigma(e_{ij}^l)}{\sum_{j'\sim i} \sigma(e_{ij'}^l) + \epsilon}, \quad G_{B,ij}^l = \frac{\sigma((e_{ij}^l)^\top)}{\sum_{j'\sim i} \sigma((e_{ij'}^l)^\top) + \epsilon}, \quad (9)$$

where $W_{A,1}, W_{B,1} \cdots, W_5 \in \mathbb{R}^{h \times h}$ denote the model weights, $G_{ij}^l$ denotes the dense attention map for element-wise gating. The convolution operation integrates the edge feature to accommodate the significance of edges in routing problems. The final prediction of the edge heatmap in TSP is $\mathbf{H}_{i,j} = \text{Softmax}(\text{norm}(\text{ReLU}(W_e e_{i,j}^L)))$ for subsequent decoding and searching process.

## 5 EXPERIMENTS

### 5.1 EXPERIMENTAL SETUP

**Hardware.** MatPOENet is trained on an NVIDIA RTX3090 24GB GPU with AMD 3970X 32-Core CPU for $N \leq 50$, and on an RTX8000 48GB GPU with Intel Xeon W-3175X CPU for $N \approx 100$. MatDIFFNet is trained on 8 NVIDIA H800 80GB GPUs with Intel Xeon (Skylake, IBRS) 16-core CPU. All evaluations are conducted on a single RTX3090 GPU with AMD 3970X 32-Core CPU.

**Training data generation.** The train data cover four CO problems: non-metric Asymmetric TSP (ATSP), 2D Euclidean TSP (abbr. 2DTSP, metric and symmetric), Directed HCP (DHCP), and 3SAT, all in their general TSP matrix formulation. Then we generate the training data for different problems by the following protocols: **i) For ATSP, 2DTSP, and HCP,** we first randomly choose the number of nodes $N$ from [min_scale, max_scale]. Then **ii) For ATSP,** we generate the distance matrix $\mathbf{C}$ from the distribution in $\text{Uniform}(0,1)$ with the diagonal entries being 0; **iii) For 2DTSP,** we assign each node a random 2D coordinate by distribution $\text{Uniform}(0,1) \times \text{Uniform}(0,1)$ and then compute the Euclidean distance matrix $\mathbf{C}$; **iv) For DHCP,** we generate a node sequence $\tau = (i_1, i_2, \cdots, i_N)$ by randomly permuting all the nodes and assigning $\mathbf{C}_{i_n,i_{n+1}} = 0$ for nodes in $\tau$ and $\mathbf{C}_{i_N,i_1} = 0$, thus ensuring a Hamiltonian cycle in $\mathbf{C}$. Then we pick a random amount of node pairs $(i,j)$ and set $\mathbf{C}_{ij} = 0$ as the noise edge. Finally we set all $\mathbf{C}_{ij} = 1$ for the rest node pairs $(i,j)$. **v) For 3SAT,** the TSP instance scale $N$ is tied to the number of variables and clauses (Appendix A.2), so we cannot set $N$ to an arbitrary value. We first pick a set of variable number $N_v$ and clause number $N_c$ as specified in Appendix D.2 to ensure that the scale of the reduced TSP instance fits different experimental scales. Finally, the instances are transformed to the general TSP form, as described in Sec. 3.2.

**Testing data preparation.** Due to the absence of previous work to standardize the evaluation of matrix-formatted general TSP across various problem distributions, we prepare 10K test instances for 3 scales (conforming to mainstream works of the "multi-task" concentration, see Table 6 in Appendix B.4), comprising 2,500 instances featuring each problem (ATSP, 2DTSP, HCP and 3SAT) in the matrix formulation, in pursuit of comprehensive examination across general TSP tasks.

Table 2: Main experimental results. Reported data for ATSP and 2DTSP are tour length. "Single": models trained and tested on each problem respectively. "Mixed": unified models trained with a mixture of 4 tasks on each scale. Asterisked (*): a unified model trained with a mixture of 4 tasks and 3 scales. **Bold**: the best result of neural solvers. Underlined: the reference results for computing the optimality gap. Red / blue boxes: ours that outperform LKH with 10K/500 trials respectively. Time: the average time (seconds) per instance solving over each line, with batch size set to 1.

| | Methods | Train Data | ATSP↓ | 2DTSP↓ | DHCP (L↓, FR↑) | | 3SAT (L↓, FR↑) | | Avg. L↓ | Avg. Gap↓ | Avg. FR↑ | Time |
|---|---|---|---|---|---|---|---|---|---|---|---|---|
| Scale: $N \approx 20$ | Gurobi | - | 1.5349 | 3.8347 | 0.0000 | 100.00% | 0.0000 | 100.00% | 1.3424 | - | 100.00% | 0.135 |
| | LKH (10000) | - | 1.5349 | 3.8347 | 0.0008 | 99.92% | 0.0000 | 100.00% | 1.3426 | 0.01% | 99.96% | 0.327 |
| | LKH (500) | - | 1.5349 | 3.8347 | 0.0056 | 99.44% | 0.0000 | 100.00% | 1.3438 | 0.11% | 99.72% | 0.038 |
| | Nearest Neighbor | - | 2.0069 | 4.5021 | 3.8556 | 0.48% | 3.0504 | 0.32% | 3.3428 | 149.02% | 0.40% | 0.000 |
| | Farthest Insertion | - | 1.7070 | 3.9695 | 3.3136 | 1.76% | 4.8816 | 0.00% | 3.4679 | 158.34% | 0.88% | 0.000 |
| | MatNet | ATSP | 1.5871 | 4.2612 | 2.9608 | 1.12% | 3.4772 | 0.56% | 3.0716 | 128.82% | 0.84% | 0.005 |
| | MatNet | Mixed | 1.6359 | 3.9114 | 0.9740 | 27.60% | 3.4656 | 11.04% | 2.4967 | 85.99% | 19.32% | 0.005 |
| | MatNet-8x | Mixed | 1.5645 | 3.8478 | 0.1936 | 80.92% | 1.6272 | 1.36% | 1.8083 | 34.71% | 41.14% | 0.037 |
| | DIMES | Mixed | 2.2335 | 4.1696 | 2.9448 | 2.67% | 2.6660 | 2.12% | 3.0035 | 123.74% | 2.39% | 0.035 |
| | DIMES-AS(100) | Mixed | 1.6790 | 3.9092 | 0.4596 | 60.12% | 0.2828 | 77.12% | 1.5826 | 17.90% | 68.62% | 0.522 |
| | MatPOENet | Mixed | 1.6445 | 3.8643 | 0.8676 | 32.60% | 0.4540 | 61.88% | 1.7076 | 27.21% | 47.24% | 0.006 |
| | MatPOENet-8x | Mixed | 1.5695 | 3.8389 | 0.1760 | 82.68% | 0.0112 | 98.88% | 1.3989 | 4.21% | 90.78% | 0.043 |
| | MatPOENet*-8x | Mixed | **1.5506** | **3.8372** | **0.0556** | **94.44%** | **0.0008** | **99.92%** | **1.3610** | **1.39%** | **97.18%** | 0.043 |
| Scale: $N \approx 50$ | Gurobi | - | 1.5545 | 5.6952 | 0.0000 | 100.00% | 0.0000 | 100.00% | 1.8124 | - | 100.00% | 0.296 |
| | LKH (10000) | - | 1.5548 | 5.6953 | 0.0000 | 100.00% | 0.2784 | 74.80% | 1.8821 | 3.85% | 87.40% | 0.513 |
| | LKH (500) | - | 1.5557 | 5.6964 | 0.0000 | 100.00% | 0.4796 | 61.80% | 1.9329 | 6.65% | 80.90% | 0.059 |
| | Nearest Neighbor | - | 2.0945 | 6.9977 | 5.1120 | 0.00% | 5.9872 | 0.00% | 5.0548 | 178.90% | 0.00% | 0.000 |
| | Farthest Insertion | - | 1.8387 | 6.0998 | 4.0224 | 5.28% | 10.3964 | 0.00% | 5.5893 | 208.39% | 2.64% | 0.001 |
| | MatNet | ATSP | 1.5753 | 7.3618 | 1.4856 | 11.80% | 8.4020 | 0.00% | 4.7062 | 159.67% | 5.90% | 0.007 |
| | MatNet | Mixed | 1.8098 | 6.0000 | 0.9288 | 30.84% | 1.1900 | 30.52% | 2.4821 | 36.95% | 30.68% | 0.007 |
| | MatNet-8x | Mixed | 1.7340 | 5.8664 | 0.3056 | 71.52% | 0.2992 | 73.08% | 2.0513 | 13.18% | 72.30% | 0.064 |
| | GLOP | Single | 1.8885 | 6.6499 | 3.7244 | 0.84% | 4.9816 | 0.76% | 4.3111 | 137.87% | 0.80% | 0.115 |
| | DIMES | Mixed | 2.3341 | 6.6271 | 3.1788 | 1.56% | 5.3656 | 0.12% | 4.3764 | 141.47% | 0.84% | 0.055 |
| | DIMES-AS(100) | Mixed | 1.6920 | 5.9447 | 0.5908 | 46.04% | 1.5830 | 20.44% | 2.4528 | 35.33% | 33.24% | 2.016 |
| | MatPOENet-8x | Single | **1.5643** | **5.7042** | 0.0652 | 93.52% | 0.1888 | 81.72% | **1.8806** | **3.76%** | 87.62% | 0.066 |
| | MatPOENet | Mixed | 1.6881 | 5.7694 | 0.1444 | 86.20% | 1.3644 | 27.08% | 2.2416 | 23.68% | 56.64% | 0.009 |
| | MatPOENet-8x | Mixed | 1.6417 | 5.7283 | **0.0172** | **98.28%** | 0.2456 | 77.60% | 1.9082 | 5.29% | 87.94% | 0.067 |
| | MatPOENet*-8x | Mixed | 1.6285 | 5.7575 | 0.0280 | 97.20% | 0.1172 | 88.44% | 1.8828 | 3.88% | **92.82%** | 0.067 |
| | MatDIFFNet | Single | 2.0713 | 5.7954 | 2.0992 | 15.32% | 0.0464 | 98.16% | 2.5031 | 38.11% | 56.74% | 0.157 |
| | MatDIFFNet-2OPT | Single | 1.7186 | 5.7279 | 0.8324 | 44.08% | **0.0188** | **98.64%** | 2.0744 | 14.46% | 71.36% | 0.165 |
| | MatDIFFNet | Mixed | 1.8385 | 6.2332 | 2.0648 | 15.76% | 0.1112 | 94.68% | 2.5619 | 41.35% | 55.22% | 0.155 |
| | MatDIFFNet-2OPT | Mixed | 1.6591 | 5.8619 | 0.8192 | 44.52% | 0.0496 | 95.64% | 2.0975 | 15.73% | 70.08% | 0.164 |
| Scale: $N \approx 100$ | Gurobi | - | 1.5661 | 7.7619 | 0.0000 | 100.00% | 0.0000 | 100.00% | 2.3320 | - | 100.00% | 0.689 |
| | LKH (10000) | - | 1.5674 | 7.7709 | 0.0000 | 100.00% | 1.0008 | 44.80% | 2.5848 | 10.84% | 72.40% | 0.811 |
| | LKH (500) | - | 1.5704 | 7.8015 | 0.0000 | 100.00% | 1.6656 | 28.08% | 2.7594 | 18.33% | 64.04% | 0.095 |
| | Nearest Neighbor | - | 2.1321 | 9.6696 | 5.4016 | 0.20% | 8.3236 | 0.00% | 6.3859 | 173.84% | 0.10% | 0.002 |
| | Farthest Insertion | - | 1.9333 | 8.4847 | 3.1256 | 26.64% | 23.5160 | 0.00% | 9.2649 | 297.29% | 13.32% | 0.003 |
| | MatNet | ATSP | **1.6217** | 19.0644 | 17.8620 | 0.00% | 40.1188 | 0.00% | 19.6667 | 743.34% | 0.00% | 0.015 |
| | MatNet | Mixed | 1.9849 | 8.2551 | 0.9776 | 31.68% | 2.0408 | 13.84% | 3.3146 | 42.14% | 22.76% | 0.018 |
| | MatNet-8x | Mixed | 1.9210 | 8.1028 | 0.3640 | 69.60% | 0.7740 | 50.76% | 2.7904 | 19.66% | 60.18% | 0.095 |
| | GLOP | Single | 1.8491 | 8.8849 | 2.7850 | 2.00% | 6.4280 | 0.08% | 4.9868 | 113.84% | 1.04% | 0.176 |
| | DIMES | Mixed | 2.5186 | 9.5777 | 3.8064 | 1.16% | 3.8064 | 0.00% | 6.0418 | 174.52% | 0.58% | 0.124 |
| | DIEMS-AS(100) | Mixed | 1.6968 | 8.3390 | 0.8480 | 24.00% | 2.8040 | 7.16% | 3.4220 | 46.74% | 15.58% | 8.437 |
| | MatPOENet | Mixed | 1.9183 | 8.2987 | 0.0984 | 90.28% | 1.0704 | 32.32% | 2.8465 | 22.06% | 61.30% | 0.017 |
| | MatPOENet-8x | Mixed | 1.8655 | 8.1719 | 0.0052 | 99.48% | 0.2440 | 77.12% | 2.5717 | 10.28% | **88.30%** | 0.094 |
| | MatPOENet*-8x | Mixed | 1.7607 | 8.0817 | **0.0012** | **99.88%** | 0.3244 | 70.92% | 2.5420 | **9.01%** | 85.40% | 0.095 |
| | MatDIFFNet | Single | 1.9432 | 7.9684 | 4.4536 | 2.96% | 0.0404 | 98.44% | 3.6014 | 54.43% | 50.70% | 0.103 |
| | MatDIFFNet-2OPT | Single | 1.7165 | **7.8482** | 1.1404 | 37.72% | **0.0240** | **98.60%** | 2.6823 | 15.02% | 68.16% | 0.112 |
| | MatDIFFNet | Mixed | 1.8763 | 8.9030 | 3.2524 | 5.68% | 0.1940 | 90.52% | 3.5564 | 52.50% | 48.10% | 0.102 |
| | MatDIFFNet-2OPT | Mixed | 1.6965 | 8.1804 | 0.9148 | 43.04% | 0.0952 | 91.44% | 2.7217 | 16.71% | 67.24% | 0.114 |

**Metrics.** **Found Rate (FR):** the percentage of optimal solutions found in the test instances for the decisive problems HCP and SAT. A higher found rate indicates better performance. We also report the the average FR (**Avg. FR**) of DHCP and 3SAT. **Tour Length (L):** This is the conventional metric for any TSP. The lower length indicate the better performance of the general TSP solver. We report the average tour length (**Avg. L**) over all the instances of different problem distributions. The average gap (**Avg. Gap**) is the performance drop from Gurobi w.r.t the mean tour length across the four tasks.

**Compared methods.** **Exact solver: Gurobi**. Note that the current Concorde only fits Euclidean TSP. **Heuristics: LKH** (500 and 10,000 trials), **Nearest Neighbor**, and **Farthest Insertion**. **Neural sovlers: Vanilla MatNet** (Kwon et al., 2021), **DIMES** (Qiu et al., 2022) (Trained under UniCO. DIMES-AS($T$) means tuned heatmap by $T$ steps of active search, detailed in Appendix E.1), **GLOP** (Ye et al., 2024b) (using vanilla MatNet as local reviser under GLOP framework). ATSP results of **BQ-NCO** (Drakulic et al., 2023) and **GOAL** (Drakulic et al., 2024) are put in Appendix F.7 for reference only (estimated by their reported optimality gap as no pre-trained checkpoints are available). **MatPOENet** (ours) and **MatDIFFNet** (ours) w/ and w/o post-inference improvements (8x parallel running for MatPOENet and 2OPT for MatDIFFNet). All evaluated neural methods are re-trained and tested on our unified dataset for fair comparison. Parameter settings are listed in Appendix D.1. Note that for HCP and SAT problems, previous specific models performed on totally

Table 3: Results of both solving time and solving quality comparing Mat-X-Net (ours) and different settings of LKH. LKH-N: LKH with 1 *runs* and N *max_trials*. Batch size = 1.

| | Method | Time↓ | ATSP↓ | 2DTSP↓ | DHCP (L↓, FR↑) | | 3SAT (L↓, FR↑) | | Avg. L↓ |
|---|---|---|---|---|---|---|---|---|---|
| $N \approx 50$ | MatPOENet | 7m31s | 1.6417 | 5.7283 | 0.0172 | 98.28% | 0.2456 | 77.60% | 1.9082 |
| | MatDIFFNet | 21m48s | 1.7186 | 5.7279 | 0.8324 | 44.08% | 0.0188 | 98.64% | 2.0744 |
| | LKH-500 | 9m9s | 1.5557 | 5.6964 | 0.0000 | 100.00% | 0.4796 | 61.80% | 1.9329 |
| | LKH-1000 | 13m53s | 1.5554 | 5.6957 | 0.0000 | 100.00% | 0.4160 | 65.68% | 1.9168 |
| | LKH-10000 | 1h22m | 1.5548 | 5.6953 | 0.0000 | 100.00% | 0.2784 | 74.80% | 1.8821 |
| $N \approx 100$ | MatPOENet | 15m35s | 1.8655 | 8.1719 | 0.0052 | 99.48% | 0.2440 | 77.12% | 2.5717 |
| | MatDIFFNet | 28m21s | 1.7165 | 7.8482 | 1.1404 | 37.72% | 0.0240 | 98.60% | 2.6823 |
| | LKH-500 | 15m49s | 1.5704 | 7.8015 | 0.0000 | 100.00% | 1.6656 | 28.08% | 2.7594 |
| | LKH-1000 | 24m39s | 1.5692 | 7.7909 | 0.0000 | 100.00% | 1.4400 | 32.52% | 2.7000 |
| | LKH-10000 | 2h15m | 1.5674 | 7.7709 | 0.0000 | 100.00% | 1.0008 | 44.80% | 2.5848 |

Table 4: Case study of $N > d$: $d_1 = 512, d_2 = 32$ on the scale $N \approx 50$.

| Method | ATSP↓ | 2DTSP↓ | DHCP (L↓, FR↑) | | 3SAT(L↓, FR↑) | | Avg. L↓ |
|---|---|---|---|---|---|---|---|
| MatPOENet ($N << d_1$) | 1.6417 | 5.7283 | 0.0172 | 98.28% | 0.2456 | 77.60% | 1.9082 |
| MatPOENet ($N > d_2$) | 1.8799 | 5.9742 | 0.4548 | 60.24% | 0.3292 | 75.56% | 2.1595 |

different problem formulation, thus orthogonal to our target of general matrix encoding TSP solving pipeline and are reasonably excluded from evaluation in this paper. Detailed clarification of this issue is sincerely explained in Appendix B.6 for possible concerns.

## 5.2 RESULTS AND DISCUSSIONS

Main results are shown in Table 2. In the following part, RQ1-RQ3 discuss the eternal topics of performance, efficiency, and scalability; RQ4 analyzes the ablations upon the POE embedding; RQ5-RQ7 introduce different applications and additional experiments based on UniCO.

**RQ1: Performance.** MatPOENet outperforms LKH when $N \approx 50$ and 100 on the average length of the full dataset. Outperforming LKH on TSP has been a pursuit by neural CO solver for long, while scarcely are there methods achieving it, let alone on the problem of general TSP. As highlighted in Table 2, on the scale $N \approx 50$, MatPOENet* achieves to outperform LKH with max_trials=500 on the Avg. L i.e., the overall performance, and on the scale $N \approx 100$, MatPOENet* outperforms LKH with as many trials as max_trials=10,000 on the Avg. L metric. *The competent results compared with LKH indicates that our method MatPOENet* can not only serve as a strong baseline for future research, but also becomes a current SOTA over the problem of general TSP.* Diving deeper into the results, we observe that LKH performs stably well on general TSP instances in continuous data space (ATSP and 2DTSP) but is not always that good in discrete data space: It may be because of the crash of the inherent so-called "alpha-measure" of LKH, which relies on the computing of minimum 1-tree that is not unique in the discrete cases. We detail the observation and analyses for LKH in Appendix G.1. The discovery of LKH's crash also suggests the significance of developing neural solvers for general TSP that can work simultaneously on the instances in both continuous and discrete spaces.

**RQ2: Efficiency.** We extract the total solving time in Table 3 for a clearer comparison. On $N \approx 50$ data, MatPOENet achieves better tour length (**1.91** v.s. 1.93) within shorter time (**7m31s** v.s. 9m9s) compared to LKH-500. More impressively, on $N \approx 100$ data, MatPOENet not only beats LKH-500 on both run-time efficiency and average quality, but also show superiority in solving performance (**2.57** v.s. 2.58) compared to LKH-10000 which consumes **8.7x** time (**15m35s** v.s. 2h15m). MatDIFFNet consumes more solving time for its complex inference denoising steps of diffusion, yet also reaches competitive results against LKH within a similar sovling time.

**RQ3: Scalability.** Experiments are conducted from three aspects to show the multi-scale general-izability: **i) MatPOENet trained on all the scales.** We train an MatPOENet with training data of all the scales, denoted by MatPOENet* in Table 2. Competitive results are obtained compared to single training, demonstrating the feasibility of, and even benefits brought by multi-scale training. **ii) Scale-free initializer POE on the scale $N > d$.** In response to the motivation of designing POE in Sec. 4.2.1, we show that MatPOENet can be trained on instances at the scale $N$ larger than the pre-set one-hot embedding dimension $d$, which is the case where vanilla MatNet fails. Results in the case $N > d$ are given in Table 4. Compared with the model trained with a higher $d$, we observe that a reduced $d$ would cause a reasonable degeneracy of model performance. **iii) On larger-scaled data.** In terms of solving large scaled CO problems, resorting to divide-and-conquer paradigm is popular

Table 5: Results on large-scaled ATSP instances.

| Scale | Method | ATSP (L↓) | Time | BiTSP (L↓) | Time | HCP (L↓) | FR↑ | Time |
|-------|--------|-----------|------|------------|------|----------|-----|------|
| 1K | Greedy | 2.146 | 21s | 5.609 | 21s | 5.734 | 0.00% | 21s |
| | MatNet | 2.130 | 1m24s | 4.352 | 1m | 3.063 | 0.78% | 45s |
| | Ours | **2.092** | **57s** | **0.517** | **40s** | **0.563** | **64.84%** | **41s** |
| 10K | Greedy | 9.516 | 4m32s | 5.938 | 4m32s | 5.438 | 0.00% | 4m32s |
| | MatNet | 9.184 | 23m17s | 3.313 | 6m10s | 2.750 | 0.00% | 23m22s |
| | Ours | **8.355** | **5m27s** | **0.688** | **5m10s** | **0.625** | **62.50%** | **5m40s** |

and proved feasible and performant (Ye et al., 2024b; Luo et al., 2024; Fu et al., 2021), where a strong solver at small-to-medium instances is still of irreplaceable importance. We readily incorporated the state-of-the-art GLOP (Ye et al., 2024b) which greatly improves MatNet's scalability. Results of the GLOP-empowered MatPOENet (Ours) are presented below, which can be regarded as an initial indication for the efficiency of our proposed POE at large scaled problems (128 instances for $N = 1K$ and 16 instances for $N = 10K$). Note BiTSP is a direct simulation of decisive CO problems in general TSP formulation, referring to randomly generated instances that has binary distances but doese not necessarily possess a zero-length loop. A more thorough discussion of scalability and applicability issues from the view of current research status of unified NCO is deferred to Appendix G.2.

**RQ4: Ablations of POE (Appendix F.1). i) MatNet v.s. MatPOENet.** Main results in Table 2 validate the effectiveness of introducing POE to enhance MatNet. **ii) Comparison of different initial rough solver.** Results of nearest neighbor (NN), farthest insertion (FI), and an ablation without rough solvers (Non) in Table 10 show superiority of NN as the initial solver for POE. We speculate that it is because NN pays more attention to the local structure, complementing subsequent attention layers where global information are learned. **iii) POE v.s. alternatives.** Fixed random vectors and trainable embeddings are tested as the substitution of the proposed POE. Results in Table 11 show that our POE outperforms the other two options suggested in the paper of MatNet (Kwon et al., 2021).

**RQ5: UniCO for pretraining-Finetuning (Appendix F.2). With MatPOENet**, UniCO executes both **i) cross-task** and **ii) cross-scale** experiments for pretraining-finetuning applications. **With MatDIFFNet**, UniCO also trains the model in the curriculum learning manner, i.e., models on larger scales are trained upon existing weights gained during training of small scaled data. Results in Appendix F.2 demonstrate applicability of UniCO as a pretraining-finetuing framework.

**RQ6: UniCO for other learning paradigms and problems. (Appendix F.3 & F.4).** Besides afore-mentioned methods (DRL, Generative, etc.), UniCO is also available for the mainstream supervised learning (SL) of neural heatmap for general TSP solving. We perform minimum modification to the backbone model in MatPOENet to obtain a heatmap predictor via pure supervised learning. Results are provided in Appendix F.3 Also, supplementary experiments on the vertex cover problem (VC) have been conducted using UniCO. Results and discussions are in F.4. To conclude, UniCO along with our Mat-X-Nets possess good adaptability to embrace new learning paradigms and tasks.

**RQ7: UniCO for standard TSP and real-world generalization (Appendix F.5 & F.6).** In order to reach better alignment to previous works that focused on 2DTSP only, we have conducted experiments on the standard test set shared by a series typical works (Kool et al., 2018; Joshi et al., 2019; Min et al., 2024; Qiu et al., 2022; Li et al., 2023b; Hudson et al., 2021), with 1280 instances each for TSP-50 and TSP-100. Also, we test MatPOENet and MatDIFFNet on 45 selected real-world instances ($N \in [14, 195]$) from the well-kown TSPLIB. Results (in Table 18, 19, 21, respectively) demonstrate that our designed networks manage to solve the conventional Euclidean TSPs without coordination as input, yielding comparable performance and capability of generalization towards distribution shifts.

## 6  CONCLUSION AND FUTURE WORK

**Conclusion.** In this paper, we first propose the UniCO framework to embed different CO problems in a consistent general TSP matrix format via problem reduction. Further, we delve into the under-explored realm of matrix-formulated TSP and devise MatPOENet and MatDIFFNet as plugs-in of UniCO, in aid of current lack of neural capability targeting general TSP.

**Limitations and Future Work.** We have made an initial step towards the general neural CO solver. Future works include: **i) More problems.** More CO problems reducible to general TSP (Appendix E.2) shall be included for evaluation. **ii) Stronger capability.** Gaps still exist comparing to exact solvers at specific problem solving. **iii) Better scalability.** MatDIFFNet has the potential to scale larger and work as a consistency model in the future (Appendix G.4).

ETHICS STATEMENT

The methods proposed in this paper aim to improve the field of neural combinatorial optimization (NCO), especially neural solvers with better generality. A dataset comprising synthetic instances of general TSPs with different problem distributions is released to facilitate the under-studied branch of matrix-formulated TSPs, thereby standardizing the evaluation of unified NCO solvers. To our best knowledge, no ethical issues or harmful insights of this work need to be otherwise stated.

REPRODUCIBILITY STATEMENT

The hardware and the preparation of the used data are described in Sec. 5.1. The detailed parameterization and implementation of the models for training and testing are provided in Appendix D. Source code and datasets can be accessed at `https://github.com/Thinklab-SJTU/UniCO`.

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

# Appendix

CONTENTS

# A    DETAILS OF PROBLEM REDUCTION AND SOLUTION TRANSFORMATION

For easy reading, here we give a review of the mathematical definitions of covered problems. Then we give the proof to problem reduction relations in the following sections. Note that methods discussed or proposed in this paper, are anchored to CO problems on graphs, especially those technically reducible or structurally similar to general TSP, i.e., feasibly characterizable via matrix formulation.

**Definition 1** (Traveling Salesman Problem (TSP)). *Given a complete, directed or undirected graph without self-loops denoted by $G = (\mathcal{V}, \mathcal{E})$ ($\mathcal{V} = \{1, 2, \cdots, N\}$: the node set, $\mathcal{E}$: the edge set) along with a cost matrix $\mathbf{C}$ of the shape $N \times N$ where the entry $\mathbf{C}_{ij}$ is the cost for edge $(i, j) \in \mathcal{E}$, the problem is to find the tour $\tau = (i_1, \cdots, i_N)$ to minimize the cost $\sum_{k=1}^{N-1} \mathbf{C}_{i_k i_{k+1}} + \mathbf{C}_{i_N i_1}$.*

**Definition 2** (Hamiltonian Cycle Problem (HCP)). *Given a directed or undirected graph $G = (\mathcal{V}, \mathcal{E})$, the problem is to determine whether there exists a Hamiltonian cycle in $G$.*

**Definition 3** (Boolean Satisfiability Problem (SAT) in conjunctive normal form (CNF)). *SAT aims to determine the existence of an interpretation that satisfies a given Boolean formula. A Boolean formula in CNF is represented by a conjunction (denoted by $\wedge$) of clauses that are disjunctions (denoted by $\vee$) of variables. For example, $(x_1 \vee \neg x_2) \wedge (\neg x_1 \vee x_2 \vee \neg x_3)$ is a SAT instance of two clauses $(x_1 \vee \neg x_2)$ and $(\neg x_1 \vee x_2 \vee \neg x_3)$, and three variables $x_1$, $x_2$ and $x_3$.*

A special case of SAT in CNF where the number of variables is no more than 3 is named as 3-Satisfiability (3SAT). Note any SAT problem can be reduced to 3SAT in polynomial time (Fouh et al., 2014). So without losing generality, we pick 3SAT for evaluation.

## A.1    HCP V.S. TSP

To transform the graph $G = (\mathcal{V}, \mathcal{E})$ of an HCP instance to the cost matrix $\mathbf{C}$ of a TSP instance, we simply set $\mathbf{C}_{ij} = 0$ if $(i, j) \in \mathcal{E}$ and $\mathbf{C}_{ij} = 1$ otherwise. Then the problem of finding a Hamiltonian cycle becomes finding a tour of length $0$ (also the minimum length) for the TSP instance with cost matrix $\mathbf{C}$. It is obvious that the procedure is polynomial-time.

## A.2    3SAT V.S. HCP

### A.2.1    REDUCTION FROM 3SAT TO HCP

Suppose we have a 3SAT instance of $N_v$ variables $\{x_0, x_1, \cdots, x_{N_v-1}\}$ and $N_c$ clauses, reducing it to an HCP instance follows the below two steps. More details of illustrations and proofs for claims in this subsection can be found in references, Section 28.12 of OpenDSA (Fouh et al., 2014) (slightly different).

**Step 1. Construct variable-clause nodes and edges.** We first construct $2N_c$ nodes for each variable, $2N_vN_c$ nodes in all. We call these nodes "variable-clause nodes". We assign the indices $0$ to $2N_vN_c - 1$ to these nodes. For a variable $x_i$, the indices of corresponding nodes are from $2N_ci$ to $2N_c(i + 1) - 1$. Then we construct edges $(m, m + 1)$ and $(m + 1, m)$ for $m$ ranges from $2N_ci$ to $2N_c(i + 1) - 2$. If $i < N_v - 1$, we further construct four edges $\big(2N_ci, 2N_c(i + 1)\big)$, $\big(2N_ci, 2N_c(i + 2) - 1\big)$, $\big(2N_c(i + 1) - 1, 2N_c(i + 1)\big)$, and $\big(2N_c(i + 1) - 1, 2N_c(i + 2) - 1\big)$; else if $i = N_v - 1$, we construct four edges $(2N_ci, 0)$, $(2N_ci, 2N_c - 1)$, $\big(2N_c(i + 1) - 1, 0\big)$, and $\big(2N_c(i + 1) - 1, 2N_c - 1\big)$.

**Step 2. Construct clause nodes and edges.** We then construct $N_c$ nodes for the clauses. We call these nodes 'clause nodes'. We denote the clauses as $\{C_1, C_2, \cdots, C_{N_c-1}\}$ We assign indices from $2N_vN_c$ to $2N_vN_c + N_c - 1$. If clause $C_j$ contains variable $x_i$, then we construct edges $\big(2N_ci + 2j, 2N_vN_c + j\big)$ and $\big(2N_vN_c + j, 2N_ci + 2j + 1\big)$. If clause $C_j$ contains variable $\neg x_i$, then we construct edges in the opposite directions, i.e., $\big(2N_vN_c + j, 2N_ci + 2j\big)$ and $\big(2N_ci + 2j + 1, 2N_vN_c + j\big)$.

So far, we have reduced the 3SAT instance to the graph of a HCP instance $G$ containing $2N_vN_c + N_c$ nodes in polynomial time. It can be easily proved that the 3SAT is satisfiable if and only if $G$ has a Hamiltonian path.

### A.2.2 TRANSFORM HCP SOLUTION TO 3SAT SOLUTION

If the tour $\tau$ traverse two variable-clause nodes whose index are $2N_c i + 2j$ and $2N_c i + 2j + 1$ in the order $2N_c i + 2j \rightarrow$ a clause node $\rightarrow 2N_c i + 2j + 1$, then the variable $x_i$ is set to True; else if the order is $2N_c i + 2j + 1 \rightarrow$ a clause node $\rightarrow 2N_c i + 2j$, then $x_i$ is set to False.

## B ADDITIONAL RELATED WORK

Apart from the literature mentioned in the main context, we hereby present a detailed summary of related works including conventional methods for TSP solving, machine learning methods for TSP solving, recent pursuit for a unified framework towards general combinatorial optimization, and specific solvers for HCP and SAT problem.

### B.1 CONVENTIONAL SOLVERS

**Exact Solvers.** Solvers for linear programming and mixed integer linear programming, including Gurobi (Gurobi Optimization, 2021) and CPLEX (Studio, 2020), can be used to solve general TSP with optimal solutions as output. Concorde is a famous TSP solver but is only applicable to 2D symmetric TSP (Hahsler & Hornik, 2007), and according to our preliminary trials, few open-source implementations produce stable and efficient solutions to ATSP. These methods can be time-consuming, especially in real-world applications where the scale of instances may be large.

**Heuristic Solvers.** Heuristic algorithms include nearest neighbor (NN), furthest insertion (FI), etc. Among the heuristics, LKH (currently LKH-3.0.9 (Helsgaun, 2017)) is the most famous one for its high efficiency and near-optimal solutions. Technical details of LKH has been elaborated in the main context.

### B.2 LEARNING METHODS FOR TSP

The learning models for TSP solving generally receive input features from the instance graph, where typically the node features indicates the 2D coordinates of the nodes, and edge features indicating the weight of the edges. These neural methods can be generally categorized into two classes, i.e., autoregressive (AR) and non-autoregressive (NAR) methods, according to their learning and inference paradigm.

**NAR Methods.** Non-autoregressive models usually output neural heatmaps in a one-shot manner, indicative of the predicted likelihood of each eadge being included in the optimal solution. These networks are mostly developed on the basis of GNN or its variants. GCN (Joshi et al., 2019), Att-GCN (Fu et al., 2021) are representative works that adopts graph convolutional network (Kipf & Welling, 2016) and graph representaion learning (Xiong et al., 2024) for edge prediction using supervised solution proximity. UTSP (Min et al., 2024) proposes a unsupervised framework based on Scattering Attention GNN (SAG) (Min et al., 2021). DIMES (Qiu et al., 2022) devises a novel meta-reinforce learning framework to work cooperatively with the active search technique for scaled 2D-TSP instances. The physics-inspired GNN (Schuetz et al., 2022) presents to leverage the Quadratic Unconstrained Binary Optimization (QUBO) and Ising models from statistical physics to encode optimization problems as differentiable loss functions to support scalable unsupervised learning, but the design of Hamiltonian formulation of specific problem can be difficult. Authors of Xia et al. (2024) also demonstrate the limitations of heatmap generation for TSP. Deviating from the prediction of a single solution with supervised learning, generative modeling methods (Sun & Yang, 2023; Li et al., 2023b; Hottung et al., 2021a) endeavor to characterize a distribution of high-quality solutions for a given instance. Solutions can be established by sampling from the distribution. GNARKD (Xiao et al., 2024) obtains NAR VRP solvers through knowledge distillation, but limited its evaluation to symmetric TSP and CVRP.

**AR Methods.** Instead of global prediction in one shot, sequence models decompose the TSP task into an n-step node prediction, where each step offers the prediction map of the next node selection based on the current state. The sequence models generally follow an attention based encoder-decoder model where the encoder produces embeddings of all input nodes and the decoder produces the

per-step node predictions accordingly. RL4CO community develops a comprehensive repository for this category of methods (Berto et al., 2023). Representatively, AM (Kool et al., 2018) first proposes using transformer-based attention model for routing problems. POMO (Kwon et al., 2020) proposes the policy optimization with multiple optima for reinforcement learning; Sym-NCO (Kim et al., 2022) leverages symmetricity for neural combinatorial optimization, which is widely adopted in subsequent literature. Recent work BQ-NCO (Drakulic et al., 2023) introduces a novel Markov Decision Process (MDP) formulation for COPs, leveraging bisimulation quotienting to enhance out-of-distribution robustness.

**Divide-and-Conquer Frameworks**   In the context of neural solvers for combinatorial optimization, scalability has been a longstanding concern and challenge for the community. For NAR methods, we notice that acquiring supervision for edge regret is extremely time-consuming, making it impractical for solving larger-scale problems than TSP-500. Meanwhile, RL-based sequence models also face challenges on larger-scale problems due to issues like sparse rewards and training instability, which also struggle to support training on larger-scale instances (even impractical for $N \geq 100$). To address this issue, apart from effort to tailor specific architectures or training techniques, resorting to divide-and-conquer is proven feasible and performant. Authors of Fu et al. (2021) train a small-scale model, which could be repetitively used to build heat maps for TSP instances of arbitrarily large size. Luo et al. (2024) proposes a novel self-improved learning method for better scalability of neural combinatorial optimization, powered by an innovative local reconstruction approach that iteratively generates better solutions by itself as pseudo-labels to guide efficient model training. Most recently, GLOP (Ye et al., 2024b) learns to partition large routing problems into TSPs and TSPs into Shortest Hamiltonian Path Problems (SHPPs), which subsequently get conquered by local revisers. These methods improve the scales of TSPs applicable to neural solvers up to 10000 nodes. UDC (Zheng et al., 2024b) develops a unified neural divide-and-conquer framework with a Divide-Conquer-Reunion (DCR) training method for solving general large-scale CO problems, which tries to address the sub-optimal dividing policy via high-efficiency GNNs for global instance dividing and a fixed-length sub-path solver for conquering divided sub-problems.

**Remarks.** It should be highlighted that the effective divide-and-conquer frameworks are indeed orthogonal to our proposed generic solver, where a significant synergy can be reached to enhance both model scalability and solving quality.

### B.3  COMBINATION OF NEURAL AND OR METHODS

In addition to end-to-end approaches for CO problems on graphs (COPG), resorting to the combination of neural networks and operations research (OR) methods is studied in several works. VSRLKH (Zheng et al., 2021) and NeuroLKH (Xin et al., 2021) combines the strong traditional heuristic Lin-Kernighan-Helsgaun (LKH) for TSP with reinforcement learning and supervised learning respectively. GNNGLS (Hudson et al., 2021) and NeuralGLS (Sui et al., 2023) present hybrid approaches for solving the TSP based on GNNs and Guided Local Search (GLS), which searches for solutions from neurally predicted regret of including each edge. da Costa et al. (2020), Sui et al. (2021) and Ma et al. (2023) are similar works that solve TSP through automatically learning effective 2-, 3- and k-opt heuristics in a data-driven manner. DPDP (Kool et al., 2022) proposes a Deep Policy Dynamic Programming (DPDP) scheme, which aims to combine the strengths of learned neural heuristics with those of DP algorithms. However, it only scales up to 100 nodes in terms of solving TSPs and VRPs. DeepACO (Ye et al., 2024a) is a meta-heuristic algorithm which leverages deep reinforcement learning to automate heuristic designs. GFACS (Kim et al., 2024) develops a neural-guided probabilistic search algorithm for solving COPs.

**Remarks.** A common limitation of these methods is the over dependence of their performance on the effectiveness of the local search, rendering the necessity of the neural parts questionable.

### B.4  MULTI-TASK CO MODELS

Recent literature show growing interest of general neural CO solver for multiple problems. While similar concepts, e.g., "multi-task solver", "universal solver", etc., have frequently appeared in some latest works, they often denote quite different approaches and functionalities. For example, MTCO (Li & Liu, 2023) devises a "multi-task" CO framework which first measures the similarity between

Table 6: Comparison of different recent works that involve "multi-task" or "general TSP" solving. "Multi-Task": Whether the method aims for multi-task CO solving. "Multi-Scale": Whether the model adapts to differently scaled instances on the fly without pre-setting parameters. "Single Solver": Whether a unified solver learns multiple problems simultaneously. "General TSP": Whether the method supports general TSP input, i.e., (probably asymmetric or discrete) distance matrix only. "QP": Quadratic Programming. "RL": Reinforcement Learning. "SL": Supervised Learning. "Evaluated Scale": the first line means the maximal (TSP/VRP) scale in the main experiments, and the second line (if exists) means the maximal scale evaluated in additional experiments (supplementary results, divide-and-conquer application, generalizability, etc.). "✔*": limited to variants of VRPs.

| Method | Evaluated Problems | Applicable Problems | Evaluated Scale | Multi-Task | Multi-Scale | Single Solver | General TSP | Solver Type |
|---|---|---|---|---|---|---|---|---|
| MTCO (Li & Liu, 2023) | PFSP | | N/A | ✗ | ✔ | ✗ | ✗ | QP |
| ASP (Wang et al., 2024) | TSP, CVRP | N/A | 100 (∼300) | N/A | N/A | N/A | ✗ | ML-based |
| MAB-MTL (Wang & Yu, 2023) | TSP, CVRP, OP, KP | | 100 | ✔ | ✔ | ✗ | ✗ | RL |
| GCNCO (Li et al., 2025a) | TSP, CVRP, OP, KP | | 100 | ✔ | ✔ | ✔ | ✗ | RL |
| MVMoE (Zhou et al., 2024) | 16 VRPs | VRP variants | 100 (1000) | ✔* | ✔ | ✔ | ✗ | RL |
| MTNCO (Liu et al., 2024) | 11 VRPs | VRP variants | 100 (∼1000) | ✔* | ✔ | ✔ | ✗ | RL |
| UNCO (Jiang et al., 2024) | TSP, CVRP, KP, MVCP, SMTWTP | | 100 | ✔ | ✔ | ✔ | ✗ | LLM + RL |
| GOAL (Drakulic et al., 2024) | ATSP, CVRP, OP, JSSP | | 100 (1000) | ✔ | ✔ | ✔ | ✔ | SL |
| MatNet (Kwon et al., 2021) | ATSP, FFSP | $\mathcal{P}$ in matrix format | 100 | ✗ | ✗ | ✗ | ✔ | RL |
| DIMES (Qiu et al., 2022) | TSP, MIS | | 10K | ✗ | ✗ | ✗ | ✗ | Meta-RL |
| T2T & Fast T2T (Li et al., 2023b; 2024) | TSP, MIS | | 10K | ✗ | ✔ | ✗ | ✗ | Generative |
| BQ-NCO (Drakulic et al., 2023) | TSP, ATSP, CVRP, OP | | 100 (1000) | ✗ | ✗ | ✗ | ✔ | RL |
| GLOP (Ye et al., 2024b) | TSP, ATSP, CVRP, PCTSP | (Large) VRPs | 10K (100K) | ✗ | ✔ | ✗ | ✔ | RL |
| **UniCO** (Ours) | ATSP, TSP, HCP, SAT | $\mathcal{P} \leq_P$ general TSP or $\mathcal{P}$ in matrix format (VC, Clique, VRPs, FFSP, MIS, etc.) | 100 (10K) | ✔ | ✔ | ✔ | ✔ | RL, Meta-RL, SL, Generative, etc. |

CO problems, and then transfers knowledge between similar instances within the *same* problem type to gain search speed-ups in its quadratic programming solver. However, it primarily focuses on the permutation flowshop scheduling problem (PFSP) and does not explore the application of its method in handling different types of problems. Moreover, MTCO is not specifically designed for learning-based neural solvers, making the term "multi-task" in its title misleading from a learning perspective and irrelevant to the theme of this paper. ASP (Wang et al., 2024) proposes a "universal" framework to address generalization issues of neural CO solvers, which iteratively improves the generalizability to different distributions (including scales). ASP proposes a model-/problem-agnostic training policy to improve generalizability. However, it is also focused on PFSP, and does not provide a specific neural solver which we believe is technically nontrivial that can deal with different types of problems. In the work (Wang & Yu, 2023), authors propose a multi-armed bandit framework to train a neural solver for different CO problems, whereby problems share a common encoder but differ in the header and the decoder. Yet, as the header and decoder necessitate customized designs for specific problems, the solver cannot be readily applied to unseen problems, restricting its applicability to the training dataset comprising TSP, CVRP, OP and KP. MVMoE (Zhou et al., 2024) develops a multi-task VRP solver with mixture-of-experts utilizing a hierarchical gating mechanism, which achieves good zero-shot generalization performance on multiple VRP variants. Boisvert et al. (2024) proposes a new generic representation that encodes problem constraints into a graph structure by breaking down each constraint into an abstract syntax tree and connecting related variables and constraints through edges. However, the authors also identify limitations in terms of training time and

the size of the generated graphs. Resembling MVMoE, MTNCO (Liu et al., 2024) tackles the cross-problem generalization among variants of VRPs using shared underlying attributes and solve them simultaneously via a single model through attribute composition. More recently, UNCO (Jiang et al., 2024) resorts to large language models (LLMs) that take natural language to formulate text-attributed instances for different COPs and encode them in the same embedding space, thereby facilitating a unified process of solution construction. But the solving quality and scalability has considerable room for improvement. GOAL (Drakulic et al., 2024) proposes a single backbone plus light-weight problem specific adapters that solves a variety of COPs include ATSP but with inferior performance.

### B.5 SPECIFIC SOLVERS FOR SAT AND HCP

**SAT.** Neural methods for solving the SAT problem can be broadly classified into two categories: standalone neural solvers and neural-guided solvers. Direct neural solvers, such as NeuroSAT (Selsam et al., 2018) and subsequent works (Cameron et al., 2020; Jaszczur et al., 2020), classify CNF formulas as satisfiable or unsatisfiable while simultaneously constructing possible assignments by decoding literal embeddings. Several alternative approaches (Amizadeh et al., 2019a;b; Ozolins et al., 2022) focus on directly generating satisfying assignments, leveraging different GNN architectures and employing unsupervised loss for training. These methods generally aim to predict a single satisfying solution for each instance, failing to account for other possible solutions. In the category of neural-guided methods, NeuroCore (Selsam & Bjørner, 2019) and #Neuro (Vaezipoor et al., 2021) utilize neural networks to guide the branching decisions of SAT and #SAT solvers. NSNet (Li & Si, 2022) models satisfiability problems as probabilistic inference, using a graph neural network (GNN) to parameterize belief propagation (BP) in the latent space, thereby guiding a local search for a satisfying assignment. CryptoANFNet (Zheng et al., 2024a) proposes a graph structure based on Arithmetic Normal Form (ANF) to more effectively encode cryptographic problems and uses GNN to solve the corresponding SAT instances. Methods have also been proposed using neural models to generate pseudo-industrial SAT instances to resolve data bottlenck (Li et al., 2023a; Chen et al., 2024). Readers can refer to Guo et al. (2023) for a thorough survey of solving SAT problem with neural approaches.

**HCP.** It is noteworthy that HCP have not yet been extensively discussed within the ML4CO community. One recent work incorporating HCP as a case study is Wang et al. (2021), which proposes a bi-level framework with an upper-level learning method to optimize the graph (e.g., adding, deleting, or modifying edges), combined with a lower-level heuristic algorithm solving the optimized graph. This framework utilizes an actor-critic-based RL method to train a graph convolutional network (GCN) and tests it on 1001 large HCP instances, achieving results comparable to LKH.

### B.6 NOTE ON EVALUATION OF SPECIFIC SOLVERS

It can be controversial that we have not yet incorporated the so-called "specialized solvers" for the individual type of CO problems covered in this paper. We hereby provide our detailed consideration and clarification upon this issue in hope to eliminate possible ambiguity and misunderstanding.

**Conforming to Our Motivation.** As our primary motivation goes, proposing a competitive yet compact workflow capable of effectively tackling (A)TSP instances with different problem distributions comes foremost.

**Lack of Companion Methods.** To our best knowledge, few existing specialized solvers match our formulation of CO problems, nor are there prevalent evaluation protocols. Notably, there is limited exploration of the potential to unify various combinatorial optimization problems on graphs and exploit matrix representations for developing neural solvers with cross-task universality or cross-distribution robustness.

**Existing Research Convention.** There is precedent in top literature for evaluating different problems within a primarily targeted problem type. E.g., in NeuroSAT (Selsam et al., 2018), the authors, despite focusing on a new SAT solver, gained recognition for its cross-task applicability by modeling and solving graph coloring, dominating-set, and node cover problems within the SAT formulation.

**Remarks.** Therefore, it is supposed to be reasonable to compare within the scope of general TSP solvers on our unified matrix datasets where various CO tasks are implicitly embedded, also in hope to facilitate standardized comparison of the general neural combinatorial optimization community.

## C  NETWORK DETAILS

### C.1  VANILLA MATNET AND MATPOENET

**Encoder.** Each node $i$ in the graph form of the input matrix is assigned with two embedding vectors[1] $\mathbf{a}_i, \mathbf{b}_i \in \mathbb{R}^d$. We denote the embedding matrix of all nodes at the $l$-th layer as $\mathbf{A}^l = [\mathbf{a}_1^l, \mathbf{a}_2^l, \cdots, \mathbf{a}_N^l] \in \mathbb{R}^{d \times N}$ and $\mathbf{B}^l$ of the same shape. The input $\mathbf{A}^0$ and $\mathbf{B}^0$ are initialized as zero embeddings and one-hot embeddings respectively. Note that $\mathbf{B}^0$ may not be a square matrix, so MatNet forces $d \geq N$ and set $\mathbf{B}^0$ as an one-hot square matrix $\mathbf{OE} \in \mathbb{R}^{N \times N}$ padded with zero values $\mathbf{0} \in \mathbb{R}^{N \times (d-N)}$, i.e., mathematically $\mathbf{B}^0 = [\mathbf{OE}, \mathbf{0}]^\top$. Notably, $\mathbf{B}^0 = \mathbf{POE}$ in MatNetPOE and need not paddings.

Taking embedding $\mathbf{A}$ as example:

$$\mathbf{Q}_a^l = \mathbf{W}_a^Q \mathbf{A}^l, \ \mathbf{K}_a^l = \mathbf{W}_a^K \mathbf{B}^l, \ \mathbf{V}_a^l = \mathbf{W}_a^V \mathbf{B}^l, \tag{10}$$

$$\mathbf{MixedScoreAtt}_a^l = \text{softmax}\left(\text{MLP}_1\left(\left[\mathbf{C}; \frac{\mathbf{Q}_a^{l\top}\mathbf{K}_a^l}{\sqrt{d_{qkv}}}\right]\right)\right), \tag{11}$$

$$\mathbf{A}^{l+1} = \text{MLP}_2\left(\mathbf{MixedScoreAtt}_a^l \mathbf{V}_a^{l\top}\right)^\top, \tag{12}$$

where $\mathbf{W}_a^Q, \mathbf{W}_a^K, \mathbf{W}_a^V \in \mathbb{R}^{d_{qkv} \times d}$ are learnable parameters for attention modeling, $[\cdot; \cdot]$ denotes the concatenation operation, $\text{MLP}(\cdot)$ denotes a multilayer perceptron layer with activation functions and batch normalization operations inside. The input and output dimensions of $\text{MLP}_1$ are 2 and 1 respectively, so $\text{MLP}_1$ achieves to mix the attention values and the cost matrix, yielding the mixed score $\mathbf{MixedScoreAtt}_a^l \in \mathbb{R}^{N \times N}$. The input and output dimensions of $\text{MLP}_2$ is are both $d$.

Computing $\mathbf{B}^{l+1}$ is completely symmetrical with $\mathbf{A}^{l+1}$ as shown in Eq. 10 to Eq. 12, by exchanging the positions of $\mathbf{A}^l$ and $\mathbf{B}^l$ and introducing new parameters $\mathbf{W}_b^Q, \mathbf{W}_b^K, \mathbf{W}_b^V$.

MatNet extends the mixed-score attention to a multi-head one, just as the original Transformer (Vaswani et al., 2017).

**Decoder.** Given the output embedding $\mathbf{A}$ and $\mathbf{B}$ of the last layer of the encoder, the decoder aims to conduct the so-called 'rollout' to obtain a tour $\tau = \{i_1, i_2, \cdots, i_N\}$. The first step is to compute the key and value matrices by $\mathbf{K}_{dec} = \mathbf{W}_{dec}^K \mathbf{B}$ and $\mathbf{V}_{dec} = \mathbf{W}_{dec}^V \mathbf{B}$, where $\mathbf{W}_{dec}^K, \mathbf{W}_{dec}^V \in \mathbb{R}^{d_{qkv} \times d}$ are trainable parameters. When node $i_n$ is selected as $n$-th node in the tour $\tau$, the query vector $\mathbf{q}_{dec}^{n+1} \in \mathbb{R}^{d_{qkv}}$ containing information of previously selected nodes $\{i_1, \cdots, i_n\}$ is computed by:

$$\mathbf{q}_{dec}^1 = \mathbf{W}_{dec}^{Q^1} \mathbf{a}_{i_1}, \ \mathbf{q}_{dec}^{n+1} = \mathbf{W}_{dec}^{Q^0} \mathbf{a}_{i_n} + \mathbf{q}_{dec}^1, \tag{13}$$

where $\mathbf{W}_{dec}^{Q^1}, \mathbf{W}_{dec}^{Q^0} \in \mathbb{R}^{d_{qkv} \times d}$ are trainable parameters for the query vectors. Then, the output embedding of $n$-th iteration of the rollout $\mathbf{o}_{dec}^{n+1}$ can be obtained by:

$$\mathbf{o}_{dec}^{n+1} = \text{Linear}\left(\text{softmax}\left(\mathbf{Inf}_{i_{n'} \leq n} + \frac{\mathbf{q}_{dec}^{n\top}\mathbf{K}_{dec}}{\sqrt{d_{qkv}}}\right)\mathbf{V}_{dec}\right), \tag{14}$$

where $\mathbf{Inf}_{i_{n'} \leq n}$ is a vector of length $N$ whose $i_{n'}$-th element is set to negative infinity for all $n' \leq n$ and other elements are set to 0, $\text{Linear}(\cdot)$ is a linear layer whose input dimension is $d_{qkv}$ and output dimension is $d$. In practice, Eq. 14 is further enhanced by multi-head attention. By Eq. 14, $\mathbf{o}_{dec}^{n+1}$ becomes a linear combination of the embeddings $\mathbf{b}$s of unselected nodes, while the weights are determined by the selected nodes whose information is contain in $\mathbf{q}_{dec}^n$.

The probability vector to select the next node $i_{n+1}$ is:

$$\mathbf{p}_\theta(i_{n+1}|i_{n'\leq n}) = \text{softmax}\left(\mathbf{Inf}_{i_{n'}\leq n} + \tanh\left(\frac{\mathbf{B}^\top \mathbf{o}_{dec}^{n+1}}{\sqrt{d}}\right)\right). \tag{15}$$

After $N$ rollout iterations, a complete tour can be obtained.

---

[1] By default, all mentioned vectors are column vectors.

**Model training.** It is trained in the same way as POMO (Kwon et al., 2020) based on DRL. For each instance with $\mathbf{C}$, it samples $N$ tours $\{\tau^1, \tau^2, \cdots, \tau^N\}$ with different nodes as start. Then the policy gradient is:

$$\nabla_\theta J(\theta) \approx \frac{1}{N} \sum_{n=1}^{N} \left( L(\tau^n) - b(\mathbf{C}) \right) \nabla_\theta \log p_\theta(\tau^n | \mathbf{C}), \tag{16}$$

where $L(\tau)$ is the length of tour $\tau$, $b(\mathbf{C})$ is a baseline method which is instantialized as the mean tour length of the $N$ tours, mathematically $b(\mathbf{C}) = 1/N \sum_{n=1}^{N} L(\tau^n)$.

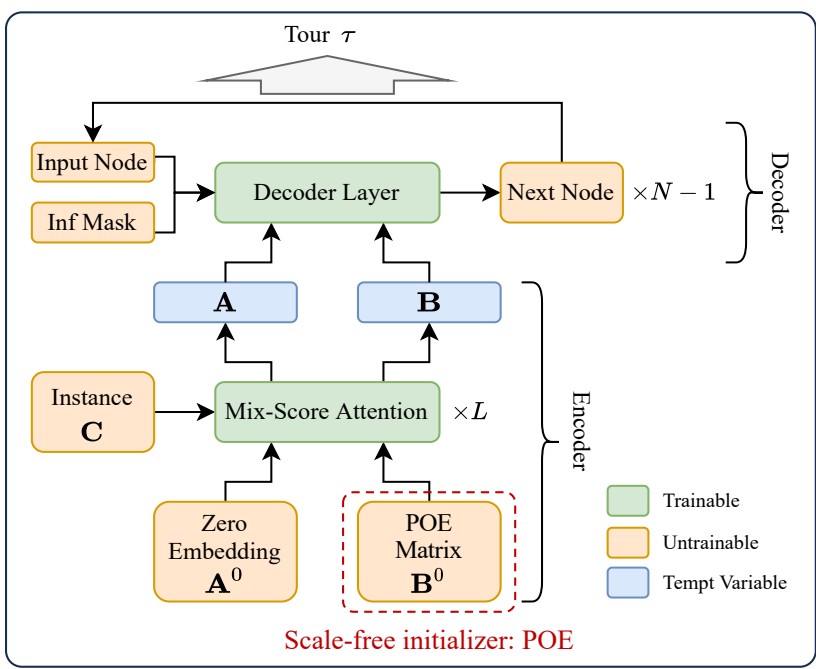

Figure 4: General structure of MatPOENet.

Fig. 4 illustrates the general neural structure of MatPOENet, highlighting with dashed square the adaptive design for initial node embedding, i.e., the scale-free initializer, pseudo one-hot embedding scheme.

## C.2    GRAPH-BASED DIFFUSION AND MATDIFFNET

**Overview.**    Given the distribution of problem instance with distance matrix $\mathbf{C}$, solutions ($\mathbf{S}$) can be established by sampling from the distribution and maximizing the conditional likelihood estimation $\mathbb{E}[\log p_\theta(\mathbf{S}|\mathbf{C})]$, where $\theta$ is the model parameters. The model is optimized through the evidence lower bound (ELBO) in Eq. 17, where $q$ is the posterior and $\mathbf{Z}$ is the latent variable.

$$\mathcal{L} = -\mathbb{E}_{q(\mathbf{Z}|\mathbf{S},\mathbf{C})} \left[ \log \frac{p_\theta(\mathbf{S}, \mathbf{Z}|\mathbf{C})}{q(\mathbf{Z}|\mathbf{S}, \mathbf{C})} \right] \geq \mathbb{E} \left[ -\log p_\theta(\mathbf{S}|\mathbf{C}) \right]. \tag{17}$$

**Details.**    MatDIFFNet generates general TSP solutions $\mathbf{S}_0 \in \{0,1\}^{n \times n}$ by $T$-step denoising process from random noises $\mathbf{S}_T$, and the latent variables include noised solution $\mathbf{S}_{1:T}$, outputting a binary heatmap guiding the search of a valid TSP tour. The discrete diffusion models generate solutions $\mathbf{S}_0 \in \{0,1\}^{n \times n}$ by $T$-step denoising process from random noises $\mathbf{S}_T$, and the latent variables include noised solution $\mathbf{S}_{1:T}$. For each entry, the model estimates a Bernoulli distribution indicating whether this entry should be selected. In implementation, each entry of solution is represented by a one-hot vector[2] such that $\mathbf{S}_0 \in \{0,1\}^{n \times n \times 2}$. Following the notations of Ho et al. (2020); Austin et al. (2021), the general framework of diffusion includes a forward noising and a reverse denoising Markov process.

---

[2] Each entry with $[0,1]$ indicates that it is included in $\mathbf{S}$ and $[1,0]$ indicates the opposite.

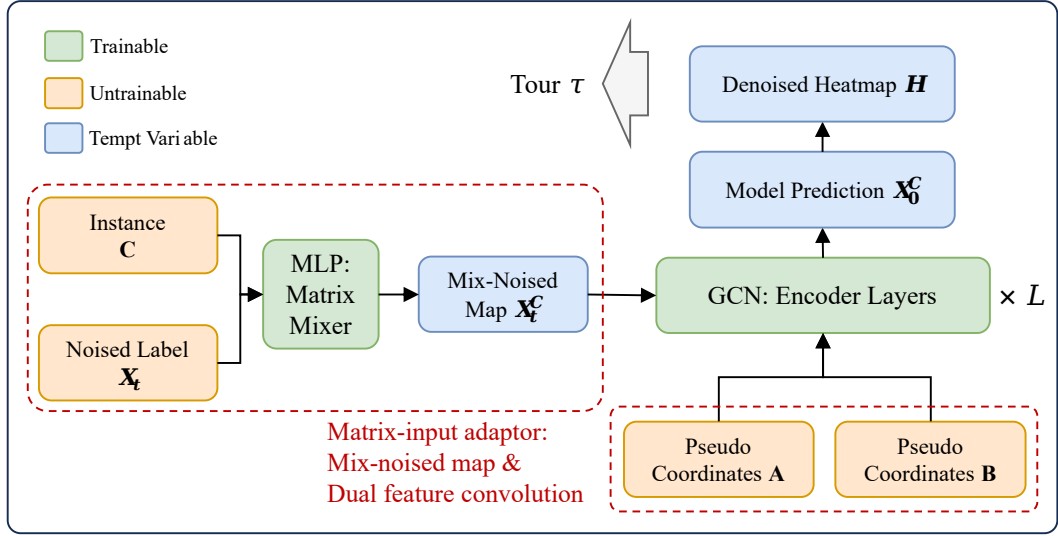

Figure 5: General structure of MatDIFFNet.

The noising process takes the initial solution $\mathbf{S}_0$ and progressively introduces noise to generate a sequence of latent variables $\mathbf{S}_{1:T}$. The denoising process is learned by the model, which starts from the final latent variable $\mathbf{S}_T$ and denoises $\mathbf{S}_t$ at each time step to generate the preceding variables $\mathbf{S}_{t-1}$ based on the instance $\mathbf{C}$, eventually recovering the target data distribution. The formulation of the denoising process is expressed as $p_\theta(\mathbf{S}_{0:T}|\mathbf{C}) = p(\mathbf{S}_T) \prod_{t=1}^{T} p_\theta(\mathbf{S}_{t-1}|\mathbf{S}_t, \mathbf{C})$. The training optimization aims to align $p_\theta(\mathbf{S}_0|\mathbf{C})$ with the data distribution $q(\mathbf{S}_0|\mathbf{C})$ using ELBO:

$$\mathcal{L} = \mathbb{E}_q \left[ \sum_{t>1} D_{KL} \left[ q(\mathbf{S}_{t-1}|\mathbf{S}_t, \mathbf{S}_0) \| p_\theta(\mathbf{S}_{t-1}|\mathbf{S}_t, \mathbf{C}) \right] - \log p_\theta(\mathbf{S}_0|\mathbf{S}_1, \mathbf{C}) \right] + C. \tag{18}$$

Specifically, the forward noising process is achieved by multiplying $\mathbf{S}_t \in [0, 1]^{N \times N \times 2}$ at step $t$ with a forward transition probability matrix $\mathbf{Q}_t \in [0, 1]^{2 \times 2}$ where $[\mathbf{Q}_t]_{i,j}$ indicates the probability of transforming $E_i$ in each entry to $E_j$. We set $\mathbf{Q}_t = \begin{bmatrix} \beta_t & 1 - \beta_t \\ 1 - \beta_t & \beta_t \end{bmatrix}$ (Austin et al., 2021), where $\beta_t \in [0, 1]$ such that the transition matrix is doubly stochastic with strictly positive entries, ensuring that the stationary distribution is uniform which is an unbiased prior for sampling. The noising process for each step and the $t$-step marginal are formulated as:

$$q(\mathbf{S}_t|\mathbf{S}_{t-1}) = \mathrm{Cat}(\mathbf{S}_t; \mathbf{p} = \mathbf{S}_{t-1}\mathbf{Q}_t) \quad \text{and} \quad q(\mathbf{S}_t|\mathbf{S}_0) = \mathrm{Cat}(\mathbf{S}_t; \mathbf{p} = \mathbf{S}_0\overline{\mathbf{Q}}_t), \tag{19}$$

where $\mathrm{Cat}(\mathbf{S}; \mathbf{p})$ is a categorical distribution over $N$ one-hot variables with probabilities given by vector $\mathbf{p}$ and $\overline{\mathbf{Q}}_t = \mathbf{Q}_1\mathbf{Q}_2 \cdots \mathbf{Q}_t$. Through Bayes' theorem, the posterior can be achieved as:

$$q(\mathbf{S}_{t-1}|\mathbf{S}_t, \mathbf{S}_0) = \frac{q(\mathbf{S}_t|\mathbf{S}_{t-1}, \mathbf{S}_0)q(\mathbf{S}_{t-1}|\mathbf{S}_0)}{q(\mathbf{S}_t|\mathbf{S}_0)} = \mathrm{Cat}\left(\mathbf{S}_{t-1}; \mathbf{p} = \frac{\mathbf{S}_t\mathbf{Q}_t^\top \odot \mathbf{S}_0\overline{\mathbf{Q}}_{t-1}}{\mathbf{S}_0\overline{\mathbf{Q}}_t\mathbf{S}_t^\top}\right). \tag{20}$$

The neural network is trained to predict the logits of the distribution $\tilde{p}_\theta(\tilde{\mathbf{S}}_0|\mathbf{S}_t, \mathbf{C})$, such that the denoising process can be parameterized through $q(\mathbf{S}_{t-1}|\mathbf{S}_t, \tilde{\mathbf{S}}_0)$:

$$p_\theta(\mathbf{S}_{t-1}|\mathbf{S}_t) \propto \sum_{\tilde{\mathbf{S}}_0} q(\mathbf{S}_{t-1}|\mathbf{S}_t, \tilde{\mathbf{S}}_0)\tilde{p}_\theta(\tilde{\mathbf{S}}_0|\mathbf{S}_t, \mathbf{C}). \tag{21}$$

Fig. 5 illustrates the general neural structure of MatDIFFNet, highlighting with dashed square the adaptive design for matrix input, i.e., the mix-noised reference map and dual feature convolution scheme.

## D    EXPERIMENTAL DETAILS

### D.1    HYPERPARAMETERS

**MatPOENet.**    We set 512 as the positional embedding dimension as well as all hidden dimensions in the network. For $N \approx 20$ and 50, 8 layers of mix-score attention block is adopted to better capture problem features, whereas 5 layers are set for $N \approx 100$ for space saving. We train our model using Adam optimizer (Kingma, 2014) with a learning rate of $4 \times 10^{-4}$ with a decay of $1 \times 10^{-6}$. The batch size is set to 200 for $N \approx 20, 50$ and 150 for $N \approx 100$. Defining a training epoch as 10,000 randomly generated problem instances, we find the performance sufficiently noteworthy within 2,000 epochs for all scales, despite the fact that more training steps might produce better convergence and outcomes.

**MatDIFFNet.**    The parameter setting of MatDIFFNet basically follows Li et al. (2023b). We set the dimension of the input features for implemented graph neural networks equals 2 indicating the pseudo 2D coordinates of the nodes, and set the feature dimension of the intermediate layers to 256. The output channel dimensions of the networks are set as 2 for the classification modelling. Default number of GNN layers is set to 12. Models at all scales are trained with a cosine learning rate schedule starting from $4 \times 10^{-4}$ and ending at 0. For each type of problem distribution, we generally use 1.28M random instances for each epoch of training and train the models for 100 epochs. We apply curriculum learning and initialize the models from $N \approx 50$ checkpoints. For mixed data training, 400 epochs are conducted to gain an equivalent period to those trained individually. Training batch-size is set to 128 for $N \approx 50$ and 32 for $N \approx 100$. For the diffusion part, We implement the model with 50 inference steps for denoising, and the models are trained with 1000 denoising steps, i.e., $T_{denoise} = 1000$. We additionally apply the technique of denoising diffusion implicit models (DDIMs (Song et al., 2020)) for accelerating inference and solution reconstruction.

**UniCO-DIMES.**    We set 8 layers for the backbone GNN with 64 hidden units for each. For both outer updates and inner meta-steps, we use AdamW (Loshchilov & Hutter, 2018) as optimizer, with a learning rate of $1 \times 10^{-3}$ and $1 \times 10^{-1}$ respectively. For all scales of problems, we train the model for 1000 outer epochs, each containing $T_{meta} = 15$ inner meta-updates with $K = 1000$ samples. In the testing phase, we set the (inner) learning rate of active search to 0.5 for faster optimization and with a typical 100 steps.

### D.2    SCALE SETTING OF 3SAT INSTANCES

To align with our experimental setup ($N \approx 20, 50, 100$), we manually tailor the number of variables $N_v$ and clauses $N_c$ as outlined in Table 7. During training at a specific scale, each batch of 3SAT instances are generated with a randomly determined parameter line within the corresponding scale group.

### D.3    TRAINING CURVES

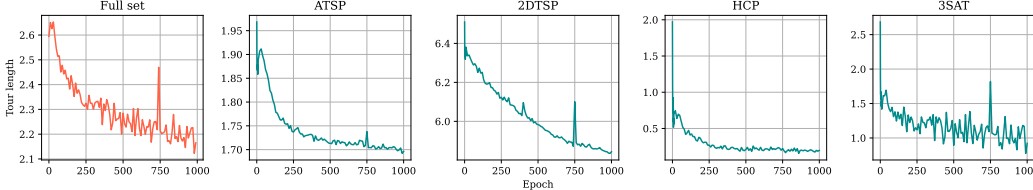

Figure 6: Tour length of different problem categories during UniCO-empowered mixed-data training of MatPOENet with $N \approx 50$.

Fig. 6, Fig. 7, and Fig. 8 depict the solved tour length during the training process of MatPOENet, MatDIFFNet, and DIMES under our UniCO framework. Note that the ATSP curve for MatDIFFNet is smoothed by exponential moving average with $\alpha = 0.01$ for better clarity, as it reaches the fluctuation stage sharply within the initial epochs. The RL-based training curves (MatPOENet and DIMES) are

Table 7: Detailed scale parameters of 3SAT instances contained in different experimental groups.

| Scale group | # Variables $(N_v)$ | # Clauses $(N_c)$ | Exact $N$ | Average $\overline{N}$ |
|---|---|---|---|---|
| $N \approx 20$ | 4 | 2 | 18 | 20.3 |
| | 3 | 3 | 21 | |
| | 5 | 2 | 22 | |
| $N \approx 50$ | 6 | 3 | 39 | 49.8 |
| | 3 | 7 | 49 | |
| | 6 | 4 | 52 | |
| | 4 | 6 | 54 | |
| | 5 | 5 | 55 | |
| $N \approx 100$ | 9 | 5 | 95 | 101.0 |
| | 5 | 9 | 99 | |
| | 8 | 6 | 102 | |
| | 6 | 8 | 104 | |
| | 7 | 7 | 105 | |

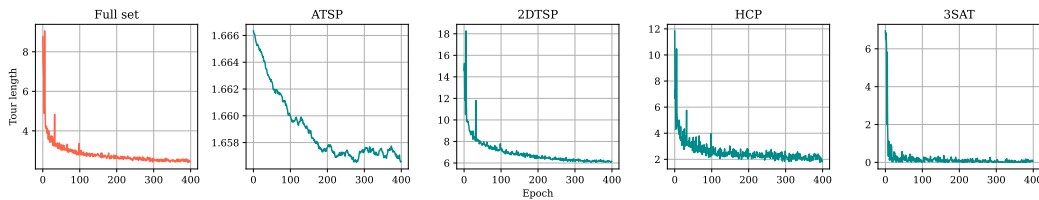

Figure 7: Tour length of different problem categories during UniCO-empowered mix-data training of MatDIFFNet with $N \approx 50$.

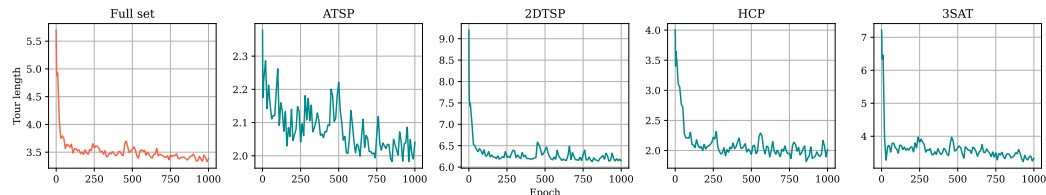

Figure 8: Tour length of different problem categories during UniCO-empowered mix-data training of DIMES-AS with $N \approx 50$.

sampled with a time step of 10. It can be observed that Transformer-, Diffusion- and GNN-based backbones are all well optimized under UniCO with problem instance reduction. Furthermore, detailed training objective curves for each individual problem type demonstrate consistency and stability of convergence across all involved problem categories, addressing our claim of multi-task robustness.

# E    MORE APPLICATIONS OF UNICO

In this section, we provide the details of possible applications of our proposed UniCO. For backbone model, we adapt DIMES (Qiu et al., 2022) for multi-task training under UniCO by simply replacing its input of nodes coordinates by random vectors. In Sec. E.1, we carefully introduce its model architecture and training policy. For more applicable problems, Sec. E.2 presents a list of feasible CO problems that can **i) be reduced to general TSP form**, **ii) formulated via matrix representation**, or **iii) resolved by our proposed models as specific solvers**. In the conclusion, we provide further clarification upon the orthogonality of UniCO and the proposed backbone solvers, reiterating the motivation and position of our paper.

## E.1 MORE ADAPTABLE BACKBONE: GNN-BASED DIMES

DIMES (Qiu et al., 2022) proposes a scalable neural solver for 2DTSP based on meta-learning and deep RL.

For TSP, parallel to attention models that sequentially decode node selections, GNN-based techniques have gained prominence for predicting heatmaps in support of online searching. To this end, we endeavor to pinpoint a GNN-based architecture that aligns seamlessly with the implementation of our instance reduction-based general CO training framework. Several observations guide this exploration: **1) High cost of supervised training**. Preparing supervision for TSP is primarily time-consuming. Even worse, matrix input of general TSP can be intractable for space as well, compared to existing works that often take the convenience of storing coordinate representations only. **2) Embedding limitations**. Prevailing GNN methods, often designed for TSP as an edge classification task, heavily rely on coordinate inputs, which remains open problem. **3) Data distribution variability**. Notable divergence of data distribution exists subsequent to the reduction of TSP from diverse problem instances. This variability poses a considerable challenge for supervised training of GNN.

Our exploration of existing models unveil that meta-learning and RL-based active search strategies in DIMES resonate with our intuitive perspective. We adapt vanilla DIMES into our UniCO architecture, underscoring its proper alignment with our motivation of tackling multi-task CO problems, and in turn validating the feasibility of our proposed framework.

**Overview.** DIMES uses an Anisotropic GNN (Bresson & Laurent, 2018) as backbone to capture representations of different types of problem instances. The vanilla DIMES takes node coordinates as the input node features and the distanrces as the input edge features. To generalize DIMES to general TSP, we empirically use random numbers sampled from [0, 1] of a fixed dimension $d$ ($d = 64$ by default) as the node feature. The training process combines Model-Agnostic Meta-Learning (MAML) and DRL. The primary objective is to enhance the prediction of heatmap initialization that later yields optimal solving results after the RL-based active search in the testing phase. Following the paradigm of meta-learning, the training process of DIMES involves *outer* epochs which optimizes the backbone GNN parameters utilizing the estimated policy gradients computed with one or more *inner* meta-updates. We present the detailed mathematical formulations below.

### E.1.1 NETWORK ARCHITECTURE

The GNN layers work as the following message passing scheme:

$$\mathbf{h}_i^{l+1} = \mathbf{h}_i^l + \alpha \left[ \text{BN}\left( \mathbf{U}^l \mathbf{h}_i^l + \mathcal{A}_{j \in \mathcal{N}_i} \left( \sigma(\mathbf{e}_{ij}^l) \odot \mathbf{V}^l \mathbf{h}_j^l \right) \right) \right], \tag{22}$$

$$\mathbf{e}_{ij}^{l+1} = \mathbf{e}_{ij}^l + \alpha \left( \text{BN}(\mathbf{P}^l \mathbf{e}_{ij}^l + \mathbf{Q}^l \mathbf{h}_i^l + \mathbf{R}^l \mathbf{h}_j^l) \right), \tag{23}$$

where $\mathbf{h}_i^l$ and $\mathbf{e}_{ij}^l$ respectively denote the node embedding of node $i$ and the edge embedding of edge $(i, j)$ at the $l$-th layer, $\mathbf{U}^l, \mathbf{V}^l, \mathbf{P}^l, \mathbf{Q}^l, \mathbf{R}^l$ are learnable parameters of $l$-th layer, $\alpha(\cdot)$ is the activation function (set as SiLU (Elfwing et al., 2018) in practice), $\text{BN}(\cdot)$ is the batch normalization function.

### E.1.2 TRAINING PROCESS

Let $F_\Phi$ denote the graph neural network parametrized by $\Phi$. For each TSP graph instance $s$ reduced from the problem pool $C$, the instance-specific input to the network is its distance matrix $D_s$, and the output $\theta_s \in \mathbb{R}^{n \times n} := F_\Phi(D_s)$ acts as the initial heatmap guiding the search of a TSP tour. A higher valued $\theta_{i,j}$ indicates a higher probability for the edge from node $i$ to node $j$ to be sampled. The vanilla loss function is articulated as the expected cost of the solution for any graph in the collection:

$$L(\Phi|C) = \mathbb{E}_{s \in C} S(\theta_s) = \mathbb{E}_{s \in C} S(F_\Phi(D_s)), \tag{24}$$

where $S(\cdot)$ denotes the sampling-based baseline function (Kool et al., 2019) from a given distribution representation of predicted heatmap. Specifically, a batch of graph instances of the same problem type and scale is generated randomly from $C$ in every *outer* training epoch, within which $T$ *inner* meta-steps are taken to fine-tune the parameters on each instance by RL-based updates, referred to as active search (Bello et al., 2017; Hottung et al., 2021b). The fine-tuned parameters $\Phi_s^{(T)}$ are

computed using these gradient updates for each graph instance $s$ with a inner learning rate $\alpha$, thus having $\Phi_s^{(0)} = \Phi$ and for $1 \leq t \leq T$:

$$\Phi_s^{(t)} = \Phi_s^{(t-1)} - \alpha \nabla_{\Phi_s^{(t-1)}} L(\Phi_s^{(t-1)}|\{s\}). \tag{25}$$

Through the inner steps, we obtain the updated heatmap $\theta_s^{(T)} = F_{\Phi_s^{(T)}}(D_s)$. For each instance $s$ in the batch, $K$ solutions $\{\tau^1, \tau^2, \cdots, \tau^K\}$ are sampled on the basis of $\theta_s^{(T)}$ to calculate the reinforcement learning-estimated gradient of $\theta$:

$$\nabla_\theta \mathbb{E}_\tau[L(\tau)] = \mathbb{E}_\tau\left[(L(\tau) - S(s))\nabla_\theta \log p_\theta(\tau)\right], \tag{26}$$

where $L(\cdot)$ denotes the length of a feasible tour and $p_\theta$ is an auxiliary distribution to compute the probability of a feasible solution for TSP with random start node and chain rule factorization:

$$p_\theta(i_{n+1}|i_{n' \leq n}) = \frac{\exp(\mathbf{s}_{i_n, i_{n+1}})}{\sum_{n'=n+1}^{N} \exp(\mathbf{s}_{i_n, i_{n'}})}. \tag{27}$$

Subsequently, we optimize the performance of the graph neural network with the estimated gradient and a meta-objective, closing the outer loop:

$$L_{\text{meta}}(\Phi|C) = \mathbb{E}_{s \in C} S(\theta_s^{(T)}|s). \tag{28}$$

### E.1.3 RESULTS AND DISCUSSION

**Performance.** As demonstrated in Table 8, with our modified input node embeddings, DIMES works finely with our UniCO framework as a matrix encoder for general TSP, achieving comparable solving quality via active searching. Despite notable gap from DIMES to MatPOENet and MatD-IFFNet, DIMES shows good compatibility of UniCO and have the potential to effectively scale to larger instances, which can also be a valuable research direction in the future.

Table 8: Results of UniCO-DIMES trained on mixed problem data.

| Scale | Methods | ATSP↓ | 2DTSP↓ | DHCP (L↓, FR↑) | | 3SAT (L↓, FR↑) | | Avg. L↓ |
|---|---|---|---|---|---|---|---|---|
| 20 | DIMES | 2.2335 | 4.1696 | 2.9448 | 2.67% | 2.6660 | 2.12% | 3.0035 |
| | DIMES-AS(100) | 1.6790 | 3.9092 | 0.4596 | 60.12% | 0.2828 | 77.12% | 1.5826 |
| | DIMES-AS(200) | 1.6439 | 3.8809 | 0.4464 | 61.92% | 0.3068 | 75.16% | 1.5695 |
| 50 | DIMES | 2.3341 | 6.6271 | 3.1788 | 1.56% | 5.3656 | 0.12% | 4.3764 |
| | DIMES-AS(100) | 1.6920 | 5.9447 | 0.5908 | 46.04% | 1.5830 | 20.44% | 2.4528 |
| | DIMES-AS(200) | 1.6794 | 5.9194 | 0.5000 | 54.20% | 1.4296 | 23.36% | 2.3821 |
| 100 | DIMES | 2.5186 | 9.5777 | 3.8064 | 1.16% | 3.8064 | 0.00% | 6.4018 |
| | DIEMS-AS(100) | 1.6968 | 8.3390 | 0.8480 | 24.00% | 2.8040 | 7.16% | 3.4220 |

**Efficiency.** Table 8 lists the solving efficiency of vanilla MatNet and DIMES under our UniCO framework. MatNet benefits from its light-weight architecture and sequential decision paradigm and have better time advantage. Active search process endows DIMES with better per-instance solving performance, while consumes much more time due to its gradient-based local search.

Table 9: Solving efficiency comparison under our UniCO framework. Setting: $N \approx 50$, batch_size = 100, 2500 instances. DIMES($T$): DIMES with $T$ inner updates of active search.

| Backbone | MatNet | MatNet(8×) | DIMES | DIMES(15) | DIMES(100) |
|---|---|---|---|---|---|
| Time | 15s | 2m40s | 2m17s | 10m35s | 1h24m |

**MatNet v.s. DIMES: MatNet wins on both efficiency and model performance, but DIMES has better scalability.** We compare vanilla MatNet without augmentation and DIMES without fine-tuning, both trained with mixed data. We see that MatNet outperforms DIMES significantly. We also present the solving time of vanilla MatNet and DIMES in Table 9, which demonstrates the high efficiency of MatNet. Though MatNet wins on both efficiency and model performance, according to their official papers, DIMES can be run on TSP-10,000 (Qiu et al., 2022), but MatNet only scales to about TSP-100 (Kwon et al., 2021), indicating better scalabilty of DIMES. It is because DIMES adopts a lighter neural architecture, but MatNet has a complex rollout scheme, causing a non-negligible amount of temporal variables, thus limiting its scalability.

**DIMES v.s. DIMES-AS($T$): The trade-off between efficiency and effectiveness brought by fine-tuning on test cases.** By comparing DIMES-AS($T$) with vanilla DIMES, DIMES-AS($T$) improves DIMES significantly with fine-tuning on the test cases. However, DIMES-AS($T$) may suffer from severe scalability issue in real-world applcations. The solving time in Table 9 shows that DIMES(15) may take about 4.6 times as DIMES.

### E.2 MORE APPLICABLE PROBLEMS

#### E.2.1 VERTEX COVER

**Definition 4** (Vertex Cover Problem (VC)). *Given an undirected graph $G$ and a positive integer $k$, a vertex cover of $G$ is a set $S$ of vertices so that every edge is incident on at least one vertex of $G$. The problem is to determine whether $G$ has a vertex cover of size no more than $k$.*

**Reduction from VC to HCP.** Let the (undirected) graph of VC instance be $\mathcal{G}(\mathcal{V}, \mathcal{E})$ and the target is to determine whether there exists a vertex cover $S$ containing no more than $k$ vertices. We can construct an HCP graph $\mathcal{G}'(\mathcal{V}', \mathcal{E}')$ with $4|\mathcal{E}| + k$ vertices. Indexing the edges in $\mathcal{E}$ from 1 to $|\mathcal{E}|$, $\mathcal{G}'$ can be mathematically described as:

$$\mathcal{V}' = \{a_1, a_2, \cdots, a_k\} \cup \{[u, i, 0], [u, i, 1] | u \in \mathcal{V}, i \in \mathcal{E}\}, \tag{29}$$

where $i$ is incident on $u$.

$$\begin{aligned}
\mathcal{E}' = &\{([u, i, 0], [u, i, 1]) \,|\, [u, i, 0] \in \mathcal{V}'\} \\
&\cup \{([u, i, a], [v, i, a]) \,|\, i \in \mathcal{E}, i = (u, v), a \in \{0, 1\}\} \\
&\cup \{([u, i, 1], [u, j, 0]) \,|\, \nexists\, e \text{ s.t. } i < e < j, [u, e, 0] \in \mathcal{V}\} \\
&\cup \{([u, i, 1], a_f) \,|\, 1 \le f \le k, \ \nexists\, e \text{ s.t. } e > i, [u, e, i] \in \mathcal{V}\} \\
&\cup \{(a_f, [u, i, 0]) \,|\, 1 \le f \le k, \ \nexists\, e \text{ s.t. } e < i, [u, e, i] \in \mathcal{V}\}.
\end{aligned} \tag{30}$$

**Transform HCP Solution to VC Decision.** If there exists a Hamiltonian cycle $\tau$ in $\mathcal{G}'$, we first select all vertices in $\tau$ that connect any vertex in $\{a_1, a_2 \cdots, a_k\}$. The selected vertices are necessarily in the form of $<u, i, a>$, and all these $u \in \mathcal{V}$ constitute the vertex cover of $\mathcal{G}$ within $k$ vertices.

Note that the VC problem has been incorporated in UniCO framework and implemented for evaluation. The details of data generation and empirical results are provided in Appendix F.4.

#### E.2.2 CLIQUE PROBLEM

**Definition 5** (Clique Problem). *A clique is a (sub)graph induced by a vertex set $K$ in which all vertices are pairwise adjacent, i.e., for all distinct $u, v \in K$, $(u, v) \in E$. A clique of size $k$ is denoted as $K_k$. The clique problem is to determine whether a graph on $n$ vertices has a clique of size $k$.*

**Reduction from clique problem to SAT.** Given an clique instance, we introduce the following to for an SAT instance:

**Variables**:

$y_{i,r}$ (true if node $i$ is the $r$-th node of the clique) for $1 \le i \le n, 1 \le r \le k$.

**Clauses:**

1) For each $r$, $y_{1,r} \vee y_{2,r} \vee \ldots \vee y_{n,r}$ (some node is the rth node of the clique).

2) For each $i$, $r < s$, $\neg y_{i,r} \vee \neg y_{i,s}$ (no node is both the $r$-th and $s$-th node of the clique).

3) For each $r \neq s$ and $i < j$ such that $(i, j)$ is not an edge of $G$, $\neg y_{i,r} \vee \neg y_{j,s}$. (If there's no edge from $i$ to $j$ then nodes $i$ and $j$ cannot both be in the clique).

That's the entire formula that will be satisfiable if and only if $G$ has a clique of size $k$. As SAT problem can be transformed to 3SAT and further reduced to general TSP in polynomial time, the decision of Clique problem can be solved by UniCO-empowered solvers.

### E.2.3 INDEPENDENT SET PROBLEM

**Definition 6** (Independent Set). *An independent set is vertex set $S$ in which no two vertices are adjacent, i.e., for all distinct $u, v \in S$, $(u, v) \notin E$.*

**Definition 7** (Complement Graph). *Let $G = (V, E)$ be a graph, the complement graph of $G$, denoted as $\overline{G} = (V, \overline{E})$, is defined such that $\overline{E}$ contains all the edges not present in $G$.*

By definition of complement, $(u, v) \in E \leftrightarrow (u, v) \notin \overline{E}$. The statement that $S \subseteq V$ is an independent set in G is equivalent to the fact that $S$ induces a clique in $\overline{G}$. Therefore, an IS problem can be transformed into a Clique decision, and solved by the same reduction of Clique problem to general TSP thereafter.

### E.2.4 VEHICLE ROUTING PROBLEMS

**Solving via sub-TSPs.** While problems like CVRP cannot be directly solved by a direct transformation, the TSP solver can still be utilized. E.g, the Cluster-First Route-Second Method (Shalaby et al., 2021), solves CVRP by first clustering points and then solving each cluster as a TSP. Thus, a robust TSP solver is effective in the second phase for CVRP. Additionally, as mentioned in Ye et al. (2024b), large routing problems can be partitioned into TSPs and TSPs into Shortest Hamiltonian Path Problems (SHPPs), solidifying the significant role a good TSP solver plays targeting sub-structures of complex VRPs.

**Solving via sub-VRPs and SAT.** We also notice that in latest works like Zheng et al. (2024b), CVRP of large scales are divided into sub-CVRPs. Thus, a theoretical valid transformation of VRP to SAT problem makes a significant difference to our proposed UniCO framework. Below provides a general formulation of the VRP constraints via variables and clauses in SAT. Subsequently, we discuss the challenges of incorporating capacity or other integer constraints via conjunctions and disjunctions of Boolean variables along with our possible solution and analysis of this idea.

Consider the three basic inputs of a VRP instance, the number of nodes $N$, the number of vehicles $K$, and the pair-wise costs $d_{ij}$. The three-dimensional decision variable $x_{ijk}$ is introduced in the SAT instance, where $i, j \in \{0, 1, \cdots, N\}$ are the indices of $N$ nodes (0 denotes the central depot), and $k \in \{1, \cdots, K\}$ is the index of $K$ vehicles. $x_{ijk}$ is the Boolean variables indicating whether vehicle $k$ is assigned to travel directly from node $i$ to node $j$.

**Constraint 1: all vehicles departs from and returns to the depot exactly once.** I.e.,

$$\forall k \in \{1, ..., K\}, \sum_{j=1}^{N} x_{0jk} = \sum_{j=1}^{N} x_{j0k} = 1, \tag{31}$$

which form the following Boolean clauses. For each vehicle $k \in \{1, ..., K\}$,

$$C_1 = (x_{01k} \vee x_{02k} \vee \cdots \vee x_{0Nk}) \tag{32}$$
$$C_2 = (x_{10k} \vee x_{20k} \vee \cdots \vee x_{N0k}) \tag{33}$$
$$C_3 = \{(\neg x_{0j_1k} \vee \neg x_{0j_2k}) | \forall j_1, j_2 \in \{1, \cdots, N\}, j_1 \neq j_2\} \tag{34}$$
$$C_4 = \{(\neg x_{i_10k} \vee \neg x_{i_20k}) | \forall i_1, i_2 \in \{1, \cdots, N\}, i_1 \neq i_2\} \tag{35}$$

**Constraint 2: all customer nodes should be visited exactly once.** Mathematically,

$$\forall i \in \{1, ..., N\}, \sum_{k=1}^{K} \sum_{j=1}^{N} x_{ijk} = 1, \tag{36}$$

which form the following Boolean clauses. For each node $i \in \{1, ..., N\}$,

$$C_5 = (x_{i11} \vee \cdots \vee x_{iN1}) \vee \cdots \vee (x_{i1K} \vee x_{i2K} \vee \cdots \vee x_{iNK}) \tag{37}$$
$$C_6 = \{(\neg x_{ij_1k} \vee \neg x_{ij_2k}) | \forall j_1, j_2 \in \{1, \cdots, N\}, j_1 \neq j_2, \forall k \in \{1, \cdots, K\}\} \tag{38}$$

So far, we have built the basic constraints of general VRPs. A Boolean formula can be obtained by $\phi = \bigwedge_{i=1}^{6} C_i$ in the SAT instance to ensure feasibility of a VRP solution.

Next, when stepping into the capacity demands for CVRP ($Q$: the maximum capacity of each vehicle; $m_i$: the demand quantity of each node), the following constraint should be added.

**Constraint 3: the loading capacity of all vehicles should not exceed the limit.** Mathematically,

$$\forall k \in \{1, ..., K\}, \sum_{i=0}^{N} \sum_{j=0}^{N} m_j \cdot x_{ijk} \leq Q \tag{39}$$

This arouse difficulty for an SAT instance with its binary variables and Boolean clauses to directly formulate the complex integer constraint. One way to handle this problem is split the vehicles and customer nodes into many sub-vehicles and sub-nodes with **unit** capacity and demand, according to the quantity of $Q$ and $m_i$. Mathematically, the number of nodes and vehicles will be changed to $\sum_{i}^{N} m_i$ and $KQ$, thus the decision space of $x_{ijk}$ is expanding to $i, j \in \{(1, 2, ..., m_1), (m_1 + 1, m_1 + 2, ..., m_1 + m_2), ..., \sum_{i}^{N} m_i\}$ and $K \in \{(1, ..., Q), (Q + 1, ..., 2Q), ..., KQ\}$. Subsequently, extra constraints shall be added to guarantee the consistency among sub-nodes and sub-vehicles which belong to the same original group, i.e., mathematically, For all nodes $i, j \in (\sum_{s=1}^{T} m_s, \sum_{s=1}^{T+1} m_s], \forall T \in \{0, ..., N-1\}$ and vehicle $k \in (Qr, Q(r+1)], \forall r \in \{0, ..., K-1\}$, we add clauses $(\bigwedge\{x_{ijk}\}) \vee (\bigwedge\{\neg x_{ijk}\})$ to the SAT formula.

**Remarks.** Thus far, the constraint conditions of the CVRP are transformed into SAT form using the 3-dimensional variables $x_{ijk}$. Here, a satisfiable variable assignment corresponds to a feasible solution to the VRP instance. The obtained candidates can then be further optimized for lower total costs. This attempt facilitates the possibility of solving sub-VRPs within our UniCO framework in the first place and provides rich implication as another viable perspective of incorporating more COPs with complex constraints using Boolean formulation of SAT. We would like to reiterate that the above descriptions are an initial conception to foster as much inspiration for future research. In practice, considering 1) the heavy computational overhead of this transformation, 2) the maturity of the two-dimensional matrix solver for general TSP, and 3) SAT instances shall also be reduced to general TSP later in our framework, we still recommend readers to resort to transforming complex routing tasks into sub-TSPs, if within our framework.

### E.2.5 MATPOENET & MATDIFFNET AS PROBLEM-SPECIFIC SOLVER

In addition to the problems mentioned above that can be consistently learned and solved by reducing to general TSP, our proposed MatPOENet and MatDIFFNet can also serve as problem-specific individual solvers decoupled from UniCO, thus enabling a wider range of applicable problems, as long as specified in a matrix form of parameters quantifying the relationship between two groups of items. For instance, as evaluated in Kwon et al. (2021) and Li et al. (2023b), Flexible Flow Shop Problem (FFSP) and Maximal Independent Set (MIS) problem are readily modelled for MatPOENet and MatDIFFNet, respectively.

**Conclusion.** Application of the reduction scheme to a wider range of problems are theoretically guaranteed by the computational complexity theory, and will be further studied in our future research. Note that resorting to problem reduction for general solving of different CO problems has limitations, where some tasks with more complicated constraints or high transformation complexity are not practical. However, the orthogonality between UniCO and specific solvers should be noted, as more problems has the potential to be solved via individual modelling for our proposed neural solvers.

Table 10: Ablation studies for the rough solver of MatNet-POE. Scale $N \approx 50$. Trained on mixed data, tested with 8x augmentation.

| Methods | ATSP↓ | 2DTSP↓ | DHCP (L↓, | FR↑) | 3SAT (L↓, | FR↑) | Avg. L↓ | Avg. FR↑ |
|---------|-------|--------|-----------|------|-----------|------|---------|----------|
| Non-MatNet-POE | 1.7988 | 5.8178 | 0.4080 | 64.72% | 1.0020 | 34.76% | 2.2566 | 49.74% |
| FI-MatNet-POE | 1.8370 | 6.0962 | 0.1920 | 82.80% | 0.6060 | 55.72% | 2.1828 | 69.26% |
| MatPOENet (NN) | **1.6417** | **5.7283** | **0.0172** | **98.28%** | **0.2456** | **77.60%** | **1.9082** | **87.94%** |

# F SUPPLEMENTARY EXPERIMENTS

## F.1 FURTHER EXPERIMENTS OF POE

**Different Initial Rough Solvers.** As mentioned in RQ4 in the main context, we tested different solvers to prepare the rough solution to initialize POE. In Table 10, among nearest neighbor (NN), farthest insertion (FI) and without initial solution (Non), NN outperforms the others, which proves the efficacy of choosing the simple nearest neighbor heuristic as initial rough solver to capture local information for POE.

**Different Initial Node Embeddings.** As mentioned in RQ4 in the main context, we have additionally implemented the two advisable ways mentioned as alternative initial embeddings for MatNet in its paper (Kwon et al., 2021). Note that they are both aimed at breaking the dimension restriction as our proposed POE.

Table 11: Results of MatNet model equipped with different initial node embeddings on $N \approx 50$ dataset. "Trainable": $N_{max}$ different vectors made of learnable parameters. "Random": completely random vectors for each problem instance.

| Init. Embed. Type | ATSP↓ | 2DTSP↓ | DHCP (L↓, | FR↑) | 3SAT (L↓, | FR↑) | Avg. L↓ |
|-------------------|-------|--------|-----------|------|-----------|------|---------|
| MatNet-Random | 2.1044 | 5.8981 | 0.8560 | 33.20% | 1.1620 | 30.44% | 2.5051 |
| MatNet-Trainable | 1.8923 | 5.8157 | 0.4148 | 63.96% | 0.5012 | 58.52% | 2.1560 |
| MatPOENet (ours) | **1.6417** | **5.7283** | **0.0172** | **98.28%** | **0.2456** | **77.60%** | **1.9082** |

Through this experiment and results in Table 11, we ackowledge the potential capability of tailored trainable emeddings and anticipate further studies, though, within our time and knowledge, better performance is achieved using our proposed POE over vanilla random vectors or trainable parameter matrices of the same shape.

## F.2 PRETRAINING-FINETUNING APPLICATIONS

As claimed in the main context that our proposed UniCO serves properly as a pretraining-finetuning framework, we hereby present two supportive experiments in conventional scenarios:

**i) Cross-task scenario:** pretrained on several problem types, finetuned on an unseen problem type.

Table 12: Results of the cross-task finetuning on 3SAT. MatPOENet is pretrained under UniCO framework with dataset comprising ATSP, 2DTSP and DHCP instances of $N \approx 50$ for 2,000 epochs and subsequently finetuned on the new task of 3SAT data for (a much fewer) 500 iterations.

| Description | 3SAT (L)↓ | 3SAT(FR)↑ | #Epochs |
|-------------|-----------|-----------|---------|
| Pretrained on ATSP, 2DTSP, DHCP w/o Finetuning | 1.4080 | 17.92% | 2000 |
| Pretrained on ATSP, 2DTSP, DHCP w/ Finetuning on 3SAT | 0.0404 | 95.96% | 50 |
| | 0.0360 | 96.40% | 100 |
| | 0.0292 | 97.08% | 200 |
| Trained on merely 3SAT (control group) | 0.0400 | 96.08% | 2000 |

From Table 12, We observe that after finetuning on the unseen task data (3SAT), the model performance is improved significantly compared with that without finetuning (the first row of the table), and can yield a similar performance compared to the case where the model is premarily trained on 3SAT only (the third row of the table). These results demonstrate the applicability of UniCO for pretraining-finetuing, which also directs a feasible path to address the generalizability issue of MatNet-POE.

**ii) Cross-scale scenario:** pretrained on smaller scaled dataset, finetuned on larger scaled dataset.

Table 13: Results of the cross-scale finetuning experiments on $N \approx 100$ sets. The MatPOENet model is pretrained under UniCO framework with different dataset of $N \approx 50$ for 2,000 epochs and subsequently finetuned on mixed data of $N \approx 100$ scale for (a much fewer) 500 iterations.

| Pretrain data | Finetune data | ATSP↓ | 2DTSP↓ | DHCP (L↓, FR↑) | | 3SAT (L↓, FR↑) | | Avg. L↓ |
|---|---|---|---|---|---|---|---|---|
| Mixed-50 | - | 2.5656 | 8.5404 | 23.5064 | 0.00% | 17.5564 | 0.00% | 13.0422 |
| Mixed-50 | Mixed-100 | 1.9798 | **8.0573** | 1.7796 | 6.60% | 2.0664 | 11.68% | 3.4708 |
| ATSP-50 | Mixed-100 | **1.8855** | 8.3838 | **0.6712** | **46.72%** | 0.2888 | 72.76% | **2.8073** |
| Mixed-100 | - | 1.8655 | 8.1719 | 0.0052 | 99.48% | **0.2440** | **77.12%** | 2.5717 |

Remarkably, we observed from Table 13 that pretraining on smaller instances of ATSP leads to greatly improved outcomes when fine-tuned on larger scaled multi-task datasets, possibly suggesting the significance of capturing general problem patterns as well as handling hard cases during pretraining. The bold numbers are the best among finetuned settings, approaching competitive performance of the model primarily trained on $N \approx 100$ data (the last row) with much shorter training time.

These empirical findings provide strong support for our assertion regarding UniCO's capability to serve as a pretraining-finetuning framework. Furthermore, they spark further interest and exploration into leveraging this paradigm to enhance the model's scalability as well as generalization performance.

### F.3 COMPARISON OF LEARNING PARADIGMS

We have already tried to combine the idea of instance reduction to general TSP with **supervised** approach. To ensure fair competition, we modified MatNet architecture into a heavy-encoder and light-decoder model for SL, which by convention, outputs a heatmap to guide subsequent local searches. Note that calculating supervision solutions for ATSP/2DTSP at scale is extremely time-consuming, we implemented the method on decision problem (HCP and 3SAT) instances where the ground truth is easier to obtain.

Table 14: Performance comparison for HCP and 3SAT problems with different learning paradigms under UniCO.

| Method | HCP-50 (L↓) | FR↑ | Time | 3SAT-50 (L↓) | FR↑ | Time |
|---|---|---|---|---|---|---|
| Greedy | 6.014 | 0.04% | 0s | 5.987 | 0.00% | 0s |
| LKH | 0.000 | 100.00% | 1m29s | 0.140 | 86.28% | 2m11s |
| MatNet | 0.481 | 53.04% | 4m12s | 0.329 | 73.12% | 2m |
| UniCO-RL | 0.017 | 98.28% | 4m8s | 0.117 | 88.44% | 2m4s |
| UniCO-SL | **0.000** | **100.00%** | **2m29s** | **0.077** | **92.32%** | **2m39s** |

Table 15: Performance comparison for larger HCP and 3SAT problems with different learning paradigms under UniCO.

| Method | HCP-500 (L↓) | FR↑ | Time | 3SAT-500 (L↓) | FR↑ | Time |
|---|---|---|---|---|---|---|
| Greedy | 6.924 | 0.00% | 1m27s | 15.149 | 0.00% | 1m15s |
| LKH | 0.526 | 55.88% | 1h1m | 5.812 | 3.92% | 1h10m |
| UniCO-SL | **0.127** | **88.12%** | **1h24m** | **2.124** | **65.60%** | **1h22m** |

Results show that SL paradigm also works performantly under our UniCO framework, with better scalability but more preliminary overheads for supervision and post searching. (UniCO-RL refers to the setting where UniCO is instantiated by MatPOENet.)

## F.4 RESULTS ON VERTEX COVER PROBLEM

Echoing the introduction of more applicable problems in Appendix E.2, we have conducted additional experiments on vertex cover (with 2500 test cases on $N \approx 50$), following the reduction procedures provided. Similar to the SAT-distributed general TSP, we specified the important parameters for data generation as specified in Table 16. The experimental results are shown in Table 17.

Table 16: Parameters for vertex cover instance generation.

| TSP Scale $N$ | Num_Edges $E$ | Cover Size $k$ | Num_Nodes $N_{VC}$ | Range of $N$ | Average $N$ |
|---|---|---|---|---|---|
| $N \approx 50$ | $Uniform(10, 12)$ | $Uniform(3, 8)$ | $Uniform(8, 17)$ | $[43, 56]$ | 50.0 |

Note that our initial experimental results show that our proposed models are easily adaptable into the UniCO framework and perform decently on the new problem VC. With the transformation of complementary graphs, a series of node selection tasks on the graph could also be addressed similarly. Thus far, a representative range of edge-wise tasks (TSP, ATSP, HCP), node-wise tasks (VC, Clique, etc.) and decisive problem (SAT) combined have formed a good coverage of mainstream COPs on graphs. In the future, we will delve further into the detailed implementation and improvements on the instance-level performance and work on more problems.

## F.5 RESULTS ON STANDARD SYMMETRIC TSP

To better align our work with previous literature in pursuit of 2D TSP, additional experiments have been conducted on the consistent test dataset of symmetric TSP-50 and TSP-100. Each dataset consists of 1280 instances featured by node coordinates. Results are shown in Table 18 and Table 19. The results of previous works are re-implemented or re-executed with the provided model weights within our consistent evaluation environment to ensure fair comparison.

**Note.** All methods except MatNet and those proposed in this paper adopt 2D coordinates as input node features. In contrast, our methods solely encode the distance matrix as input information. Additionally, our models can be trained on mixed data, whereas previous works were exclusively trained on symmetric TSP only. These comparative results demonstrate that our models can achieve comparable (and even better in some cases) performance without utilizing coordinates, validating our proposed MatPOENet and MatDIFFNet as successful matrix encoders for symmetric TSP.

## F.6 RESULTS ON REAL-WORLD INSTANCES

In the main experiment, this paper primarily focuses on synthetic data for testing. To address this limitation, we have additionally tested our methods on a subset of standard TSPLIB instances, which are renowned for their real-world scenarios in TSP evaluation. We selected 45 instances with city scales ranging from 14 to 195 (as listed in Table 20). These instances encompass different distance types (such as "EUC_2D", "GEO", etc.) and representation formats (such as "coordinates", "full matrix", "diagonal matrix", etc.). To ensure a fair evaluation and optimally leverage the matrix-encoding capability of our unified framework and models, we have rewritten the selected instances

Table 17: Results on vertex cover.

| Method | Tour length ($\downarrow$) | Found rate ($\uparrow$) |
|---|---|---|
| GUROBI | 0.000 | 100.00% |
| Greedy | 7.351 | 0.00% |
| MatNet | 2.349 | 1.08% |
| MatPOENet-8x | 0.220 | 78.28% |
| MatDIFFNet-2OPT | 0.477 | 66.60% |

Table 18: Performance comparison on standard symmetric TSP-50 test set.

| Method | Tour length (↓) | Optimality gap (↓) | Time/instance |
|---|---|---|---|
| Concorde (Optimal) | 5.688 | – | 0.074s |
| AM (Kool et al., 2018) | 5.747 | 1.04% | 0.013s |
| POMO (Kwon et al., 2020) | 5.698 | 0.18% | 0.218s |
| Sym-NCO (Kim et al., 2022) | 5.738 | 0.68% | 0.077s |
| GCN (Joshi et al., 2019) | 5.776 | 1.53% | 0.008s |
| UTSP+MCTS (Min et al., 2024) | 5.818 | 2.30% | 0.063s |
| DIMES+S+AS (Qiu et al., 2022) | 5.859 | 3.01% | 2.884s |
| DIFUSCO (Sun & Yang, 2023) | 5.709 | 0.38% | 0.388s |
| T2T+GS (Li et al., 2023b) | 5.690 | 0.04% | 1.164s |
| GNNGLS-1s (Hudson et al., 2021) | 5.693 | 0.10% | 1.080s |
| MatNet-8x (Kwon et al., 2021) | 5.857 | 2.97% | 0.056s |
| MatPOENet*-8x (Ours) | 5.781 | 1.63% | 0.059s |
| MatPOENet*-128x (Ours) | 5.726 | 0.66% | 0.296s |
| MatDIFFNet-2OPT (Ours) | 5.721 | 0.59% | 0.509s |

Table 19: Performance comparison on standard symmetric TSP-100 test set.

| Method | Tour length (↓) | Optimality gap (↓) | Time/instance |
|---|---|---|---|
| Concorde (Optimal) | 7.756 | – | 0.404s |
| AM (Kool et al., 2018) | 7.951 | 2.52% | 0.026s |
| POMO (Kwon et al., 2020) | 7.883 | 1.64% | 0.205s |
| Sym-NCO (Kim et al., 2022) | 7.927 | 2.21% | 0.078s |
| GCN (Joshi et al., 2019) | 8.307 | 7.08% | 0.011s |
| UTSP+MCTS (Min et al., 2024) | 8.069 | 4.46% | 0.223s |
| DIMES+S+AS (Qiu et al., 2022) | 8.061 | 3.94% | 8.508s |
| DIFUSCO (Sun & Yang, 2023) | 7.845 | 1.14% | 0.409s |
| T2T+GS (Li et al., 2023b) | 7.788 | 0.13% | 1.198s |
| GNNGLS-1s (Hudson et al., 2021) | 7.837 | 1.05% | 1.389s |
| MatPOENet*-8x (Ours) | 8.127 | 4.78% | 0.262s |
| MatPOENet*-128x (Ours) | 7.933 | 2.28% | 0.758s |
| MatDIFFNet-2OPT (Ours) | 7.840 | 1.08% | 0.769s |

Table 20: Tested 45 TSPLIB instances of scale range: [14, 195]

| | | | | |
|---|---|---|---|---|
| att48.tsp | berlin52.tsp | ch130.tsp | ch150.tsp | eil101.tsp |
| eil51.tsp | eil76.tsp | kroA100.tsp | kroC100.tsp | kroD100.tsp |
| lin105.tsp | pr76.tsp | rd100.tsp | st70.tsp | bayg29.tsp |
| bays29.tsp | brg180.tsp | fri26.tsp | gr120.tsp | gr24.tsp |
| gr48.tsp | gr96.tsp | ulysses16.tsp | ulysses22.tsp | bier127.tsp |
| d198.tsp | kroA150.tsp | kroB100.tsp | kroB150.tsp | kroE100.tsp |
| pr107.tsp | pr124.tsp | pr136.tsp | pr144.tsp | pr152.tsp |
| rat195.tsp | rat99.tsp | u159.tsp | brazil58.tsp | dantzig42.tsp |
| gr17.tsp | gr21.tsp | hk48.tsp | swiss42.tsp | burma14.tsp |

into a consistent format of "EDGE_WEIGHT_TYPE: EXPLICIT" and "EDGE_WEIGHT_FORMAT: FULL_MATRIX", with all distances scaled to $[0, 1]$. The results are given in Table 21.

The results demonstrate that our models are also effective on problem instances with completely unseen distributions and varying sizes. We would like to kindly remind the readers that an important aspect is that we are the first to evaluate the TSPLIB instances without any initial knowledge from node coordinates. Thank you again for enhancing the completeness of our evaluation. The task of improving the generalizability towards more real-world cases are planned for future research.

Table 21: Results on TSPLIB instances.

| Method | Avg. tour length |
|---|---|
| LKH3 | 4.8622 |
| Greedy | 5.4455 |
| MatPOENet | 5.0811 |
| MatDIFFNet | 5.2744 |

### F.7 FULL EXPERIMENTAL RESULTS

For your quick reference, we present a complete version of experimental results in Table 22, supplementary to Table 2 in the main context, containing both major results and most additional results. A substantial quantity of empirical investigations are conducted to provide a comprehensive evaluation of our proposed framework and models.

## G FURTHER DISCUSSIONS

### G.1 NOTE ON LKH

We have been conducting a preliminary study on the boundary of the strong heuristic LKH when dealing with general TSP beyond the 2D Euclidean space where it holds an overwhelming advantage. LKH is a heuristic with three main components: the $\alpha$-nearest measure, node penalties, and the $k$-opt searching algorithm. It operates in the following steps:

1. Compute the $\alpha$-Nearest Measure: The $\alpha$-measure of an edge is calculated based on the length of the minimum 1-tree of a graph and the minimum 1-tree containing that edge. The $\alpha$-measures are used to specify the edge candidate set.

2. Node Penalties: LKH employs a subgradient optimization technique to obtain penalties over each node and modifies the distances.

3. Search Solutions by $k$-Opt: $k$ edges in the current tour are exchanged by another set of $k$ edges from the candidate sets to improve the tour until no more exchanges can be found.

There is a fatal flaw of LKH on binary TSP (such as the SAT-distributed TSP in our case) in the first step of computing $\alpha(i, j)$. For a discrete TSP problem instance, $\alpha(i, j)$ can only be either 0 or 1. This implies that many edges may have the same $\alpha(i, j)$ value of 0, making it difficult to distinguish effective candidates and consequently leading to suboptimal performance on such discrete instances. Moreover, unlike the generation of HCP instances in our experiments where the ones and zeros are randomly sampled from a uniform distribution and a zero-length cycle is forced in each problem matrix, the 3SAT-distributed TSP cases are more complex due to their highly structured translation from variables and clauses to the HCP distance matrix. One possible approach to address this problem is to use a learned neural mapping network to transform some difficult binary distance matrices into the more softened and thus easier space to assist LKH solving.

### G.2 NOTE ON THE SCALABILITY AND APPLICABILITY

Scalability remains one of the most frequently encountered challenges in the field of neural combinatorial optimization. The following is a conclusion of our observations and clarifications regarding the scalability issue, based on our investigations and experiments.

**Current Status of General TSP Solving.** Firstly, as demonstrated by the main experiments (Table 22), although the scale of the tested instances is not extremely large, the baseline TSP solvers (including both neural methods and strong heuristics such as LKH3) do not yield satisfactory results within an acceptable time frame. This indicates that for the relatively underexplored task of general TSP, the solvers at the current scale are still insufficiently effective, not to mention generalizing to larger scales.

**A Shared Convention.** From a peer perspective, a common phenomenon can be observed in top published literature targeting "multi-task", "general", and even "universal" combinatorial optimization, that an evaluation on problems with up to 100 nodes is generally an acknowledged convention.

Table 22: Full experimental results. Reported data for ATSP and 2DTSP are tour length. "Single": models trained and tested on each problem respectively. "Mixed": unified models trained with a mixture of 4 tasks on each scale. Asterisked (*): a unified model trained with a mixture of 4 tasks and 3 scales. "8x": representing 8 parallel trials for sequential model solving. BQ-NCO (Drakulic et al., 2023) and GOAL (Drakulic et al., 2024) reports their results on ATSP but have not been open-sourced yet. We make estimation according to their reported optimality gap for reference only. **Bold**: the best score of neural solvers in each column. Underlined: the best solved length over the full set for reference. Red box and blue box : ours that outperform LKH with max_trials=10,000 and max_trials=500 respectively. Time: the average time (seconds) per instance solving over each line, with batch size set to 1.

| | ID | Methods | Train Data | ATSP↓ | 2DTSP↓ | DHCP (L↓, FR↑) | | 3SAT (L↓, FR↑) | | Avg. L↓ | Avg. Gap↓ | Avg. FR↑ | Time |
|---|---|---|---|---|---|---|---|---|---|---|---|---|---|
| | 1 | Gurobi | - | 1.5349 | 3.8347 | 0.0000 | 100.00% | 0.0000 | 100.00% | 1.3424 | - | 100.00% | 0.135 |
| | 2 | LKH (10000) | - | 1.5349 | 3.8347 | 0.0008 | 99.92% | 0.0000 | 100.00% | 1.3426 | 0.01% | 99.96% | 0.327 |
| | 3 | LKH (500) | - | 1.5349 | 3.8347 | 0.0056 | 99.44% | 0.0000 | 100.00% | 1.3438 | 0.11% | 99.72% | 0.038 |
| | 4 | Nearest Neighbor | - | 2.0069 | 4.5021 | 3.8556 | 0.48% | 3.0504 | 0.32% | 3.3428 | 149.02% | 0.40% | 0.000 |
| | 5 | Farthest Insertion | - | 1.7070 | 3.9695 | 3.3136 | 1.76% | 4.8816 | 0.00% | 3.4679 | 158.34% | 0.88% | 0.000 |
| Scale: N ≈ 20 | 6 | MatNet | ATSP | 1.5871 | 4.2612 | 2.9608 | 1.12% | 3.4772 | 0.56% | 3.0716 | 128.82% | 0.84% | 0.005 |
| | 7 | MatNet-8x | ATSP | **1.5391** | 3.9735 | 1.5476 | 9.28% | 1.9184 | 6.08% | 2.2446 | 67.21% | 7.68% | 0.036 |
| | 8 | MatNet | Mixed | 1.6359 | 3.9114 | 0.9740 | 27.60% | 3.4656 | 11.04% | 2.4967 | 85.99% | 19.32% | 0.005 |
| | 9 | MatNet-8x | Mixed | 1.5645 | 3.8478 | 0.1936 | 80.92% | 1.6272 | 1.36% | 1.8083 | 34.71% | 41.14% | 0.037 |
| | 10 | Non-MatNet-POE | Mixed | 1.7133 | 3.8990 | 0.8400 | 33.24% | 0.5760 | 50.64% | 1.7571 | 30.89% | 41.94% | 0.005 |
| | 11 | Non-MatNet-POE-8x | Mixed | 1.6057 | 3.8695 | 0.5444 | 53.28% | 0.3344 | 67.52% | 1.6057 | 19.62% | 60.40% | 0.035 |
| | 12 | DIMES | Mixed | 2.2335 | 4.1696 | 2.9448 | 2.67% | 2.6660 | 2.12% | 3.0035 | 123.74% | 2.39% | 0.035 |
| | 13 | DIMES-AS(100) | Mixed | 1.6790 | 3.9092 | 0.4596 | 60.12% | 0.2828 | 77.12% | 1.5826 | 17.90% | 68.62% | 0.522 |
| | 14 | DIMES-AS(200) | Mixed | 1.6439 | 3.8809 | 0.4464 | 61.92% | 0.3068 | 75.16% | 1.5695 | 16.92% | 68.54% | 1.124 |
| | 15 | MatPOENet | Mixed | 1.6445 | 3.8643 | 0.8676 | 32.60% | 0.4540 | 61.88% | 1.7076 | 27.21% | 47.24% | 0.006 |
| | 16 | MatPOENet-8x | Mixed | 1.5695 | 3.8389 | 0.1760 | 82.68% | 0.0112 | 98.88% | 1.3989 | 4.21% | 90.78% | 0.043 |
| | 17 | MatPOENet* | Mixed | 1.5933 | 3.8632 | 0.4052 | 62.24% | 0.1528 | 84.88% | 1.5036 | 12.01% | 73.56% | 0.006 |
| | 18 | MatPOENet*-8x | Mixed | **1.5506** | **3.8372** | 0.0556 | 94.44% | **0.0008** | 99.92% | **1.3610** | **1.39%** | **97.18%** | 0.043 |
| | 19 | Gurobi | - | 1.5545 | 5.6952 | 0.0000 | 100.00% | 0.0000 | 100.00% | 1.8124 | - | 100.00% | 0.296 |
| | 20 | LKH (10000) | - | 1.5548 | 5.6953 | 0.0000 | 100.00% | 0.2784 | 74.80% | 1.8821 | 3.85% | 87.40% | 0.513 |
| | 21 | LKH (500) | - | 1.5557 | 5.6964 | 0.0000 | 100.00% | 0.4796 | 61.80% | 1.9329 | 6.65% | 80.90% | 0.059 |
| | 22 | Nearest Neighbor | - | 2.0945 | 6.9977 | 5.1120 | 0.00% | 5.9872 | 0.00% | 5.0548 | 178.90% | 0.00% | 0.000 |
| | 23 | Farthest Insertion | - | 1.8387 | 6.0998 | 4.0224 | 5.28% | 10.3964 | 0.00% | 5.5893 | 208.39% | 2.64% | 0.001 |
| Scale: N ≈ 50 | 24 | MatNet | ATSP | 1.5753 | 7.3618 | 1.4856 | 11.80% | 8.4020 | 0.00% | 4.7062 | 159.67% | 5.90% | 0.007 |
| | 25 | MatNet-8x | ATSP | **1.5612** | 6.9445 | 0.6036 | 49.24% | 6.3468 | 0.00% | 3.8640 | 113.20% | 24.62% | 0.061 |
| | 26 | MatNet | Mixed | 1.8098 | 6.0000 | 0.9288 | 30.84% | 1.1900 | 30.52% | 2.4821 | 36.95% | 30.68% | 0.007 |
| | 27 | MatNet | Mixed | 1.7340 | 5.8664 | 0.3056 | 71.52% | 0.2992 | 73.08% | 2.0513 | 13.18% | 72.30% | 0.007 |
| | 28 | Non-MatNet-POE | Mixed | 1.8606 | 5.8855 | 1.0392 | 27.28% | 1.1416 | 31.32% | 2.4817 | 36.93% | 29.30% | 0.007 |
| | 29 | Non-MatNet-POE-8x | Mixed | 1.7988 | 5.8178 | 0.4080 | 64.72% | 1.0020 | 34.76% | 2.2566 | 24.51% | 49.74% | 0.060 |
| | 30 | Rand-MatNet-POE | Mixed | 1.8282 | 6.0207 | 0.9380 | 30.00% | 1.4404 | 22.12% | 2.5568 | 41.07% | 26.06% | 0.006 |
| | 31 | Rand-MatNet-POE-8x | Mixed | 1.7513 | 5.8853 | 0.3096 | 71.52% | 0.4708 | 60.84% | 2.1042 | 16.10% | 66.18% | 0.063 |
| | 32 | FI-MatNet-POE-8x | Mixed | 1.8370 | 6.0962 | 0.1920 | 82.80% | 0.6060 | 55.72% | 2.1828 | 20.44% | 69.26% | 0.064 |
| | 33 | MatNet-Random-8x | Mixed | 2.1044 | 5.8981 | 0.8560 | 33.20% | 1.1620 | 30.44% | 2.5051 | 38.22% | 31.82% | 0.064 |
| | 34 | MatNet-Trainable-8x | Mixed | 1.8923 | 5.8157 | 0.4148 | 63.96% | 0.5012 | 58.52% | 2.1560 | 18.96% | 61.24% | 0.071 |
| | 35 | GLOP | Single | 1.8885 | 6.6499 | 3.7244 | 0.84% | 4.9816 | 0.76% | 4.3111 | 137.87% | 0.80% | 0.115 |
| | 36 | DIMES | Mixed | 2.3341 | 6.6271 | 3.1788 | 1.56% | 5.3656 | 0.12% | 4.3764 | 141.47% | 0.84% | 0.055 |
| | 37 | DIMES-AS(100) | Mixed | 1.6920 | 5.9447 | 0.5908 | 46.04% | 1.5830 | 20.44% | 2.4528 | 35.33% | 33.24% | 2.016 |
| | 38 | DIMES-AS(200) | Mixed | 1.6794 | 5.9194 | 0.5000 | 54.20% | 1.4296 | 23.36% | 2.3821 | 31.43% | 38.78% | 3.879 |
| | 39 | MatPOENet-8x | Single | **1.5643** | **5.7042** | 0.0652 | 93.52% | 0.1888 | 81.72% | **1.8806** | **3.76%** | 87.62% | 0.066 |
| | 40 | MatPOENet | Mixed | 1.6881 | 5.7694 | 0.1444 | 86.20% | 1.3644 | 27.08% | 2.2416 | 23.68% | 56.64% | 0.009 |
| | 41 | MatPOENet-8x | Mixed | 1.6417 | **5.7283** | 0.0172 | 98.28% | 0.2456 | 77.60% | 1.9082 | 5.29% | 87.94% | 0.067 |
| | 42 | MatPOENet* | Mixed | 1.6753 | 5.8633 | 0.2112 | 80.48% | 0.8172 | 42.40% | 2.1417 | 18.17% | 61.44% | 0.008 |
| | 43 | MatPOENet*-8x | Mixed | 1.6285 | 5.7575 | 0.0280 | 97.20% | **0.1172** | 88.44% | 1.8828 | 3.88% | **92.82%** | 0.067 |
| | 44 | MatPOENet-8x (N > d) | Mixed | 1.8799 | 5.9742 | 0.4548 | 60.24% | 0.3292 | 75.56% | 2.1595 | 19.15% | 68.00% | 0.062 |
| | 45 | MatDIFFNet | Single | 2.0713 | 5.7954 | 0.0992 | 15.32% | 0.0464 | 98.16% | 2.5031 | 38.11% | 56.74% | 0.157 |
| | 46 | MatDIFFNet-2OPT | Single | 1.7186 | 5.7279 | 0.8324 | 44.08% | **0.0188** | 98.64% | 2.0744 | 14.46% | 71.36% | 0.165 |
| | 47 | MatDIFFNet | Mixed | 1.8385 | 6.2332 | 2.0648 | 15.76% | 0.1112 | 94.68% | 2.5619 | 41.35% | 55.22% | 0.155 |
| | 48 | MatDIFFNet-2OPT | Mixed | 1.6591 | 5.8619 | 0.8192 | 44.52% | 0.0496 | 95.64% | 2.0975 | 15.73% | 70.08% | 0.164 |
| | 49 | Gurobi | - | 1.5661 | 7.7619 | 0.0000 | 100.00% | 0.0000 | 100.00% | 2.3320 | - | 100.00% | 0.689 |
| | 50 | LKH (10000) | - | 1.5674 | 7.7709 | 0.0000 | 100.00% | 1.0008 | 44.80% | 2.5848 | 10.84% | 72.40% | 0.811 |
| | 51 | LKH (500) | - | 1.5704 | 7.8015 | 0.0000 | 100.00% | 1.6656 | 28.08% | 2.7904 | 18.33% | 64.04% | 0.095 |
| | 52 | Nearest Neighbor | - | 2.1321 | 9.6696 | 5.4016 | 0.20% | 8.3236 | 0.00% | 6.3859 | 173.84% | 0.10% | 0.002 |
| | 53 | Farthest Insertion | - | 1.9333 | 8.4847 | 3.1256 | 26.64% | 23.5160 | 0.00% | 9.2649 | 297.29% | 13.32% | 0.003 |
| Scale: N ≈ 100 | 54 | MatNet | ATSP | **1.6217** | 19.0644 | 17.8620 | 0.00% | 40.1188 | 0.00% | 19.6667 | 743.34% | 0.00% | 0.015 |
| | 55 | MatNet-8x | ATSP | 1.5983 | 17.8146 | 13.5196 | 0.00% | 35.3216 | 0.00% | 17.0635 | 631.71% | 0.00% | 0.094 |
| | 56 | MatNet | Mixed | 1.9849 | 8.2551 | 0.9776 | 31.68% | 2.0408 | 13.84% | 3.3146 | 42.14% | 22.76% | 0.018 |
| | 57 | MatNet-8x | Mixed | 1.9210 | 8.1028 | 0.3640 | 69.60% | 0.7740 | 50.76% | 2.7904 | 19.66% | 60.18% | 0.095 |
| | 58 | GLOP | Single | 1.8491 | 8.8849 | 2.7850 | 2.00% | 6.4280 | 0.08% | 4.9868 | 113.84% | 1.04% | 0.176 |
| | 59 | BQ-NCO | ATSP | 1.5904 | - | - | - | - | - | - | - | - | 0.016 |
| | 60 | GOAL | ATSP | **1.5771** | - | - | - | - | - | - | - | - | 0.039 |
| | 61 | Non-MatNet-POE | Mixed | 2.0307 | 8.9929 | 1.0616 | 28.76% | 1.2944 | 26.88% | 3.3449 | 43.43% | 27.82% | 0.015 |
| | 62 | Non-MatNet-POE-8x | Mixed | 1.9800 | 8.7895 | 0.4420 | 63.60% | 0.7096 | 31.80% | 3.0728 | 31.77% | 47.70% | 0.093 |
| | 63 | DIMES | Mixed | 2.5186 | 9.5777 | 3.8064 | 1.16% | 3.8064 | 0.00% | 6.4018 | 174.52% | 0.58% | 0.124 |
| | 64 | DIEMS-AS(100) | Mixed | 1.6968 | 8.3390 | 0.8480 | 24.00% | 2.8040 | 7.16% | 3.4220 | 46.74% | 15.58% | 8.437 |
| | 65 | MatPOENet | Mixed | 1.9183 | 8.2987 | 0.0984 | 90.28% | 1.0704 | 32.32% | 2.8465 | 22.06% | 61.30% | 0.017 |
| | 66 | MatPOENet-8x | Mixed | 1.8655 | 8.1719 | 0.0052 | 99.48% | **0.2440** | 77.12% | 2.5717 | 10.28% | **88.30%** | 0.094 |
| | 67 | MatPOENet* | Mixed | 1.8107 | 8.2703 | 0.0796 | 92.12% | 1.2856 | 26.04% | 2.8616 | 22.71% | 59.08% | 0.017 |
| | 68 | MatPOENet*-8x | Mixed | 1.7607 | **8.0817** | 0.0012 | 99.88% | 0.3244 | 70.92% | 2.5420 | **9.01%** | 85.40% | 0.095 |
| | 69 | MatDIFFNet | Single | 1.9432 | 7.9684 | 4.4536 | 2.96% | 0.0404 | 98.44% | 3.6014 | 54.43% | 50.70% | 0.103 |
| | 70 | MatDIFFNet-2OPT | Single | 1.7165 | **7.8482** | 1.1404 | 37.72% | **0.0240** | 98.60% | 2.6823 | 15.02% | 68.16% | 0.112 |
| | 71 | MatDIFFNet | Mixed | 1.8763 | 8.9030 | 3.2524 | 5.68% | 0.1940 | 90.52% | 3.5564 | 52.50% | 48.10% | 0.102 |
| | 72 | MatDIFFNet-2OPT | Mixed | 1.6965 | 8.1804 | 0.9148 | 43.04% | 0.0952 | 91.44% | 2.7217 | 16.71% | 67.24% | 0.114 |

Table 6 presents a recent review of these works, ranging from the applicable problems to the evaluated scales of tasks.

**Technical Limitations.** The training of MatPOENet consumes $O(N^3)$ space ($O(N^2)$ space for the attention mechanism and $O(N)$ space for $N$ rollout iterations), making it impractical to run on large-scale instances (e.g., $N \geq 500$) on a single GPU. However, as shown in Table 5, UniCO's ability to learn larger-scaled instances can be achieved by switching to different learning paradigms. For example, a modified version of heatmap-based MatPOENet performs well on $N \approx 500$ through vanilla supervised learning. Another approach to improving the ability to solve large-scale combinatorial optimization problems is to resort to the divide-and-conquer paradigm, which is also a consensus and has been proven feasible and effective (Ye et al., 2024b; Fu et al., 2021; Luo et al., 2024). In this context, a strong solver for small-to-medium-sized instances remains of irreplaceable importance. We have readily incorporated the latest work GLOP, which significantly enhances MatNet's scalability. Moreover, the promising generative models for discrete optimization can be considered as detailed in Appendix G.4, which is also planned for our future work.

### G.3 DISCUSSION OF MATPOENET V.S. MATDIFFNET

**A Trade-off among Quality, Efficiency, and Scalability.** As shown in the main result table, MatPOENet excels at smaller instances as a strong yet compact solver with faster inference and more transparent training process but can hardly scale due to its RL nature, whereas MatDIFFNet has more potential to scale to larger cases with more accurate proximity of the solution space (especially in continuous 2DTSP and complex discrete space, e.g., 3SAT) via its generative design but consumes much longer time to train and infer due to its heavier architecture and diffusion processes.

**Comparison of Stability.** MatDIFFNet does not directly generate solutions but decodes the heatmap to obtain them. Consequently, the solution quality of MatDIFFNet is less stable than that of MatPOENet. Specifically, in the case of MatDIFFNet, if there is an error in predicting a particular node, it will directly impact the predictions of all subsequent nodes. In contrast, MatPOENet generates solutions step by step and can select better subsequent nodes based on the current state even if there is an error in a particular node. This instability also results in the relatively poor performance of MatDIFFNet when solving the Hamiltonian cycle problem (HCP).

### G.4 DISCUSSION OF GENERATIVE COMBINATORIAL OPTIMIZATION

**Strong scalability of neural backbone and sparsification.** In addition to the fact that SL-based neural networks generally outperform their RL counterparts in terms of scalability, MatDIFFNet employs a GNN as the backbone encoder, which enables convenient graph sparsification (e.g., k-nearest neighbor) on TSP. This effectively reduces computational overhead and improves inference efficiency. As demonstrated in previous research (Qiu et al., 2022), training TSP solvers with a sparsification factor of, for example 50, significantly reduces the memory and time overhead when training graphs with while maintaining competitive performance. This is also one of the crucial reasons we choose general TSP, an edge-based selection task, as the endpoint for problem reduction. In contrast, such an edge sparsification scheme cannot be adopted when learning to solve most node-based selection problems, such as vertex cover and independent set.

**Rich expressivity of learning solution space distribution.** In our investigation of recent literature on neural solving of combinatorial optimization, especially for graph problems like TSP, a notable observation is the growing attention and demonstrated performance of diffusion-based generative models (Sun & Yang, 2023; Li et al., 2023b). From DIFUSCO and T2T to our proposed MatDIFFNet and its future versions, generative models possess powerful expressive and modeling capabilities. They can learn the distribution of high-quality solutions conditioned on specific problem instances, which is beneficial for providing a good starting point or exploration direction in the solution space. In the context of combinatorial optimization, this means they can potentially generate a variety of solutions that might be closer to the optimal one, rather than being limited to a single or a few predicted solutions as in some traditional methods.

**Good prospects of consistency model and unsupervised tuning.** Moreover, the latest research on generative combinatorial optimization indicates that diffusion models can be improved through

consistency training (Song & Dhariwal, 2023). Consistency models learn to map directly from different noise levels to the optimal solution for a given instance, thereby achieving a high-quality, rapid one-step generation solution. This approach reduces the number of iterations and enhances efficiency. Therefore, we will introduce consistency models into our framework in the future to optimize the inference speed and computational efficiency of diffusion models. Also, as suggested in works (Sanokowski et al., 2023; 2024), diffusion models finely foster the applicability of unsupervised learning of combinatorial optimization.

For these reasons, we believe it appropriate to present MatDIFFNet in parallel with MatPOENet in the main context. This represents both conventional powerful autoregressive solvers using reinforcement learning and a model with wider extendability and value for further exploration. We maintain that this provides a more comprehensive view of the current trend in neural combinatorial optimization.

