# OpenReview forum: "UniCO: On Unified Combinatorial Optimization via Problem Reduction to Matrix-Encoded General TSP"
_ICLR.cc/2025/Conference — ICLR 2025 Poster_

### Official Review · Reviewer_qSht · 2024-10-31

**Soundness:** 3
**Presentation:** 3
**Contribution:** 3
**Rating:** 5
**Confidence:** 5

**Summary:**

This article focuses on multi-task CO problems, proposes a solution method that is general for several CO problems and presents two efficient solvers.

**Strengths:**

1. The Problem reduction of this paper has theoretical support.
2. The proposed methods of this paper has advantages in terms of effectiveness.

**Weaknesses:**

1. The RedCO approach proposed in this paper is not intuitively applicable to a wide range of CO problems. I think the value of multi-task CO should be reflected in its applicability to most CO problems.
2. The contribution of this paper is weak, translating these problems into a TSP is not a new idea and TSP solver is quite well developed.

**Questions:**

1. In RQ2 you show a comparison of solution times and results with the LKH method. The reason you showed efficiency in this experiment compared to LKH seems to come entirely from superior performance on the 3-SAT problem. I don't think it is fair to take Average L in this case. I am more curious as to why LKH performs poorly on the 3-SAT problem and where MatPOENet and MatDIFFNet excel in this problem. Can you provide a visual example to help me understand this result intuitively?
2. For solving efficiency, I think this paper should be compared more with Gurobi for efficiency. I am very curious about the results of this part. I also suggest you add time as reference in Table1.
3. I am having trouble understanding the specific N and d settings for variants in the ii) part of RQ3 and especially in Table 4. I need more explanation about it.
4. What is the significance of MatDIFFNet? Based on the results so far (ignoring the future work you mentioned) it looks like its lagging behind in performance and efficiency as well as training efficiency. I would suggest deleting this section or putting it in the appendix. Also The authors say in RQ7 that MatDIFFNet has the potential for more accurate solution space for larger scale instances while mentioning in the limitation that ``MatDIFFNet has the potential for direct solving of larger instances but is currently yet to be implemented.`` But I can't find any evidence for this. But I can't find any evidence for this , please explain this.

---

### Official Review · Reviewer_xiNJ · 2024-10-31

**Soundness:** 3
**Presentation:** 3
**Contribution:** 4
**Rating:** 6
**Confidence:** 5

**Summary:**

This paper presents an interesting approach to dealing with multi-task CO by transforming several CO problems into equivalent TSPs. This paper also proposes two new solvers, MatPOENet and MatDIFFNet, to solve the following TSP.

**Strengths:**

1. This article is well-written and reasonable.
2. The author has carried out abundant experiments and discussions.

**Weaknesses:**

I think this article has two major weaknesses that should be considered.
1. This paper adopts a quite special modeling approach, and its applicability is worrying. I doubt the effectiveness of reducing multi-CO problems into the general TSP form featured by distance matrices.
According to Fig. 1, you show that NP problems can be transformed into SAT, I am concerned if this part can be proved. For some problems, the transformation into TSP is itself an NP problem if you want to maintain the Found Rate ``FR`` (e.g., CVRP, as mentioned in Appendix D.2.4, ``first clustering points and then solving each cluster as a TSP`` will harm the FR), and even if it can be transformed into TSP, the time complexity of such transaction may increase dramatically.
2. Some of the experiments in this paper are not clearly described. I tried my best to find out but it is still not clear what the exact settings of the * version, single, and mixed in Table 2 are.

**Questions:**

1. MatDIFFNet is trained on 8 NVIDIA H800 80GB GPUs with Intel Xeon (Skylake, IBRS) 16-core CPU is super computational resource consuming, I am curious about the ablation experiments on computational resources.
2. The test questions in the article experiments are too limited, this article mentions applicability for ``P ≤P general TSP or P in matrix format (VC, Clique, VRPs, FFSP, MIS, etc.)`` etc., I would highly recommend to introduce evaluations on more CO problems to respond to my concerns on applicability.
3. In Line 1137, you mention that ``Also, they generally evaluated their proposed methods on no larger than 100 nodes of TSP/VRP instances, with a major emphasis of methodological innovations rather than eager pursuit of scalability at sheer engineering level.`` What means the sheer engineering level? Also, it seems that this paper also mainly focuses on no larger than 100 nodes. Please provide a clear explanation.
4. This paper uses a unique test problem design, which I think requires the authors to implement more comparative algorithms (e.g., GOAL, MVMOE) on the problems they cover for experimental validation.
5. The results in Table 5 are not sufficient to illustrate performance on larger-scaled data, I think you should provide experiments without the aid of an external process to explore whether the model has the ability to scale up. Also, this paper does experiments on scales of 20-100, and I doubt that it makes sense to compare methods that address large-scale CO problems such as GLOP.

---

### Official Review · Reviewer_rbCa · 2024-11-01

**Soundness:** 3
**Presentation:** 4
**Contribution:** 3
**Rating:** 8
**Confidence:** 4

**Summary:**

Authors propose RedCO, to unify a set of CO problems by reducing them into the general TSP form featured by distance matrices. RedCO demonstrates the potential to efficiently train a neural TSP solver using a diverse range of CO instances, and can also be adapted to specialize for specific problem types.

**Strengths:**

1. The paper is well-written and easy to understand.
2. As far as I know, this is the first study attempting to create a general framework for learning various COPs in a reduction manner.
3. The experiments conducted are thorough, and the results effectively showcase the framework's capability to handle arbitrary matrix-encoded TSPs.

**Weaknesses:**

1. The organization of the experimental section is lacking. With seven research questions (RQs) presented, the lack of clear categorization makes this part of the paper somewhat difficult to navigate.
2. The results for DIFUSCO and T2T are not included. It is noted that MatDIFFNet performs well on 3SAT problems, which is developed upon DIFUSCO and T2T.
3. While the specific problem reduction is detailed in Appendix A, it would be helpful to have a more detailed introduction to the reduction principles and the applicable COPs. Specifically:
   - What types of COPs (or what properties must COPs have) can be reduced to a general TSP?
   - What considerations should be taken into account when performing this reduction?

**Questions:**

1. Table 2 shows that MatPOENet and MatDIFFNet outperform LKH in solving 3SAT problems, but they tend to produce worse results in most other scenarios. Could you provide some explanations for this?
2. How does RedCO perform on standard TSPs (Symmetric TSPs)?
3. In line 268, the POE is based on $f(x) = 1/cosh(100x)$, can you give more introduction of the empirical function?

---

> ### Author Response · Authors · 2024-11-18
> **Author Response to Reviewer rbCa (Cont.)**
>
> (Cont. A5) **TSP-100:**
> | Method                 | Tour length (↓) | Optimality gap (↓) | Time/instance |
> | -- | -- | --| -- |
> | Concorde (Optimal)     | 7.756           | --                 | 0.404s        |
> | AM | 7.951           | 2.52%              | <0.001s       |
> | POMO  | 7.883           | 1.64%              | 0.001s        |
> | Sym-NCO| 7.927           | 2.21%              | <0.001s       |
> | GCN   | 8.307           | 7.08%              | 0.011s        |
> | UTSP+MCTS  | 8.069           | 4.46%              | 0.223s        |
> | DIMES+S+AS   | 8.061           | 3.94%              | 8.508s        |
> | DIFUSCO  | 7.845           | 1.14%              | 0.409s        |
> | T2T+GS   | 7.788           | 0.13%              | 1.198s        |
> | GNNGLS-1s    | 7.837           | 1.05%              | 1.389s        |
> | MatPOENet*-8x (Ours)   | 8.127           | 4.78%              | 0.262s        |
> | MatPOENet*-128x (Ours) | 7.933           | 2.28%              | 0.758s        |
> | MatDIFFNet-2OPT (Ours) | 7.840           | 1.08%              | 0.769s        |
>
> **Note that all the other methods except MatNet and ours, adopts the 2D coordinates as input node feature, while our methods only encode the distance matrix as input information. Also, our models can be trained on mixed data while previous works were exclusively trained on symmetric TSP only.** These comparative results demonstrate that our models can achieve comparable (and even better in some cases) performance without utilizing the coordinates, validating our proposed MatPOENet and MatDIFFNet as successful matrix encoders for symmetric TSP as well.
>
> > **Q6 (Details of POE):** In line 268, the POE is based on an empirical function, can you give more introduction of the empirical function?
>
> **A6:** Thank you for your question. The motivation behind designing POE is to address the limitation of vanilla one-hot embedding (OE) in MatNet [4] that hampers its scalability. Our inspiration stems from the concept of incorporating more information about the possible connectivity between different nodes when searching for a TSP tour. We aim to assign similar embeddings (measured by the inner product) to nodes that are likely to be connected in the solution. For this purpose, we need a rough solution (calculated by the nearest neighbor heuristic) and a periodic function from which distinct values can be sampled based on varying $pos$ (node index in the rough solution tour) and $i$ (node index in the original permutation), as specified in Equation 2 (page 6). Motivated by this process, the empirical function is determined. We found that it performs better than other alternative choices suggested by the MatNet paper, such as random node embedding and trainable embeddings.
>
> | N=50             | ATSP       | 2DTSP      | DHCP(L)    | DHCP(FR)   | 3SAT(L)    | 3SAT(FR)   | Full set(L) |
> | -- | -- | -- | ---------- | ---------- | ---------- | ---------- | -- |
> | MatPOENet       | **1.6417** | **5.7283** | **0.0172** | **98.28%** | **0.2456** | **77.60%** | **1.9082**  |
> | MatNet-Random    | 2.1044     | 5.8981     | 0.856      | 33.20%     | 1.1620     | 30.44%     | 2.5051      |
> | MatNet-Trainable | 1.8923     | 5.8157     | 0.4148     | 63.96%     | 0.5012     | 58.52%     | 2.1560      |
>
> ### References:
>
> [1] Multi-task combinatorial optimization: Adaptive multi-modality knowledge transfer by an explicit inter-task distance. (arXiv:2305.12807)
>
> [2] ASP: Learn a universal neural solver! (TPAMI 2024)
>
> [3] Efficient training of multi-task combinarotial neural solver with multi-armed bandits. (arXiv:2305.06361)
>
> [4] Matrix encoding networks for neural combinatorial optimization. (NeurIPS 2021)
>
> [5] ATTENTION, LEARN TO SOLVE ROUTING PROBLEMS! (ICLR 2019)
>
> [6] POMO: Policy Optimization with Multiple Optima for Reinforcement Learning. (NeurIPS 2020)
>
> [7] Sym-NCO: Leveraging Symmetricity for Neural Combinatorial Optimization. (NeurIPS 2022)
>
> [8] An Efficient Graph Convolutional Network Technique for the Travelling Salesman Problem. (arXiv:1906.01227)
>
> [9] Unsupervised Learning for Solving the Travelling Salesman Problem. (NeurIPS 2023)
>
> [10] DIMES: A Differentiable Meta Solver for Combinatorial Optimization Problems.
>
> [11] DIFUSCO: Graph-based Diffusion Solvers for Combinatorial Optimization.
>
> [12] T2T: From distribution learning in training to gradient search in testing for combinatorial optimization. (NeurIPS 2023)
>
> [13] Graph Neural Network Guided Local Search for the Travelling Salesperson Problem.
>
> Once again, we offer our heartfelt gratitude for your precious time and meticulous review. We sincerely hope that our responses have addressed your questions and concerns and effectively cleared up potential ambiguity or misunderstandings. We would be extremely grateful if, through your valuable reconsideration of our work, a stronger consensus could be reached regarding our contribution to the open community. We are eagerly awaiting your positive reply.
>
> Best regards,
>
> The Authors

---

### Official Review · Reviewer_WPxC · 2024-11-04

**Soundness:** 3
**Presentation:** 3
**Contribution:** 3
**Rating:** 6
**Confidence:** 3

**Summary:**

This paper proposes a unified neural solver framework called RedCO, which uses problem reduction techniques to map different combinatorial optimization (CO) problems to the general Traveling Salesman Problem (TSP) format. Two novel neural solvers, MatPOENet and MatDIFFNet, are introduced to handle matrix-encoded inputs and solve these problems efficiently. This work aims to extend neural combinatorial optimization beyond specific problem types by providing a scalable solution for problems like asymmetric TSP (ATSP), directed Hamiltonian cycle problems (DHCP), and 3-Satisfiability (3SAT).

**Strengths:**

- The introduction of MatPOENet and MatDIFFNet, which use Transformer-based and diffusion-based models, respectively, showcases the application of advanced neural network structures to solve matrix-encoded TSP problems.

- The RedCO framework offers a novel approach by unifying different combinatorial optimization (CO) problems through reduction to a general TSP format. This reduction expands the scope of neural solvers to tackle diverse problem types in a single architecture.

- RedCO's capability to handle non-metric, asymmetric, and discrete TSP instances, unlike traditional Euclidean-focused TSP solvers, significantly broadens its applicability.

- The RedCO framework is designed to incorporate various solver types, including existing methods like DIMES, showing the framework's modularity.

**Weaknesses:**

- While the framework performs well for medium-scale problems, its efficiency and feasibility for large-scale, real-world instances (e.g., with tens of thousands of nodes) are not thoroughly demonstrated or tested.

- The use of complex neural models like MatPOENet and MatDIFFNet makes it difficult to understand the inner workings and decision-making processes of these solvers. More interpretability features or case studies would be beneficial.

- The paper mainly focuses on synthetic data for testing, with limited discussion on how the models would handle real-world problem instances that could have different statistical properties.

- There is little exploration into how the proposed solvers manage noisy or incomplete data, which is common in practical applications.

- The MatDIFFNet, while powerful for certain problem types, is computationally intensive, which may hinder its use for larger instances or require additional optimization strategies.

**Questions:**

1. When converting various combinatorial optimization problems to TSP instances, are there cases where the reduction process fails or underperforms due to problem characteristics? How does the method handle these instances?

2. Given that MatDIFFNet has longer inference times due to complex diffusion steps, are there plans to optimize the model architecture or algorithm to improve inference speed and computational efficiency?

3. In multi-task training, does the interaction between different tasks lead to performance drops in any specific task? Is there a clear mechanism in the model to handle task weight allocation and interdependencies?

4. How robust are MatPOENet and MatDIFFNet when the input data contains noise or incomplete information? Were there any robustness tests conducted?

---

> ### Comment · Reviewer_WPxC · 2024-11-28
>
> Thanks for your response. I decided to increase my score to 6, and I hope to see the final version at the conference.

---

### Meta-Review · Area_Chair_tAGV · 2024-12-21

**Metareview:**

This paper proposed a "unified" neural solver for combinatorial optimization (CO) problems. Its key idea is to reduce various CO problems to general TSP, and then train a neural TSP solver to solve it. Specifically, a reinforcement learning based autoregressive model and a diffusion-based generative mode. Reviewers acknowledged the novelty of this paper, especially the idea of reducing different CO problems to a unified TSP. Meanwhile, several key limitations have also been raised, including: 1) practical value could be limited, since not all CO problems can be easily transformed to TSP, 2) lack of evaluation on large-scale problems, and 3) limited contribution since transformation among different CO is well-known. After reading the paper and all reviews, I also have doubts regarding the application value of this paper. Though theoretically different NP-hard problems can be transformed, typically it is not efficient or necessary for practical problem solving. In reality, we seldom perform reduction to solve a unified problem, and often problem-specific algorithms delivers the best performance. Nevertheless, this paper may provide some new ideas to the NCO community. I consider the reasons of acceptance outweight those of rejection, but I urge authors to have a more detailed discussion on the practical limitation of this paper.

**Additional Comments On Reviewer Discussion:**

Authors provided detailed point-to-point reponses to reviewers' comments. Most concerns are addressed, and three reviewers raised score.

---

### Decision · Program_Chairs · 2025-01-22

Accept (Poster)